# Divergence Frontiers for Generative Models: Sample Complexity, Quantization Effects, and Frontier Integrals

Lang Liu[1]     Krishna Pillutla[2]     Sean Welleck[2,3]
Sewoong Oh[2]     Yejin Choi[2,3]     Zaid Harchaoui[1]

[1] Department of Statistics, University of Washington
[2] Paul G. Allen School of Computer Science & Engineering, University of Washington
[3] Allen Institute for Artificial Intelligence

## Abstract

The spectacular success of deep generative models calls for quantitative tools to measure their statistical performance. Divergence frontiers have recently been proposed as an evaluation framework for generative models, due to their ability to measure the quality-diversity trade-off inherent to deep generative modeling. We establish non-asymptotic bounds on the sample complexity of divergence frontiers. We also introduce frontier integrals which provide summary statistics of divergence frontiers. We show how smoothed estimators such as Good-Turing or Krichevsky-Trofimov can overcome the missing mass problem and lead to faster rates of convergence. We illustrate the theoretical results with numerical examples from natural language processing and computer vision.

## 1 Introduction

Deep generative models have recently taken a giant leap forward in their ability to model complex, high-dimensional distributions. Recent advances are able to produce incredibly detailed and realistic images [34, 51, 32], strikingly consistent and coherent text [50, 66, 6], and music of near-human quality [16]. The advances in these models, particularly in the image domain, have been spurred by the development of quantitative evaluation tools which enable a large-scale comparison of models, as well as diagnosing of where and why a generative model fails [55, 38, 27, 54, 31].

*Divergence frontiers* were recently proposed by Djolonga et al. [18] to quantify the trade-off between quality and diversity in generative modeling with modern deep neural networks [54, 37, 59, 44, 49]. In particular, a good generative model must not only produce high-quality samples that are likely under the target distribution but also cover the target distribution with diverse samples.

While this framework is mathematically elegant and empirically successful [37, 49], the statistical properties of divergence frontiers are not well understood. Estimating divergence frontiers from data for large generative models involves two approximations: (a) joint quantization of the model distribution and the target distribution into discrete distributions with quantization level $k$, and (b) statistical estimation of the divergence frontiers based on the empirical estimators of the quantized distributions.

Djolonga et al. [18] argue that the quantization often introduces a positive bias, making the distributions appear closer than they really are; while a small sample size can result in a pessimistic estimate of the divergence frontiers. The latter effect is due to the *missing mass* of the samples, causing the two distributions to appear farther than they really are because the samples do not cover some parts of the distributions. The first consideration favors a large $k$, while the second favors a small $k$.

35th Conference on Neural Information Processing Systems (NeurIPS 2021).

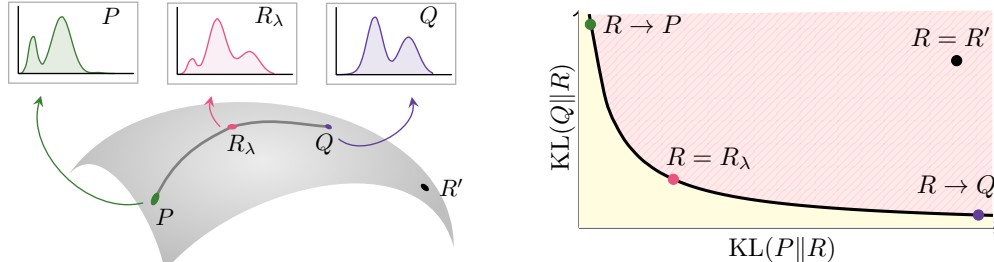

Figure 1: **Left**: Comparing two distributions $P$ and $Q$. Here, $R_\lambda = \lambda P + (1 - \lambda)Q$ is the interpolation between $P$ and $Q$ for $\lambda \in (0, 1)$ and $R'$ denotes some arbitrary distribution. **Right**: The corresponding divergence frontier (black curve) between $P$ and $Q$. The interpolations $R_\lambda$ for $\lambda \in (0, 1)$ make up the frontier, while all other distributions such as $R'$ must lie above the frontier.

In this paper, we are interested in answering the following questions: (a) Given two distributions, how many samples are needed to achieve a desired estimation accuracy, or in other words, what is the sample complexity of the estimation procedure; (b) Given a sample size budget, how to choose the quantization level to balance the errors induced by the two approximations; (c) Can we have estimators better than the naïve empirical estimator.

**Outline.** We review the definitions of divergence frontiers and propose a novel statistical summary in Sec. 2. We establish non-asymptotic bounds for the estimation of divergence frontiers in Sec. 3, and discuss the choice of the quantization level by balancing the errors induced by the two approximations. We show how smoothed distribution estimators, such as the add-constant estimator and the Good-Turing estimator, improve the estimation accuracy in Sec. 4. Finally, we demonstrate in Sec. 5, through simulations on synthetic data as well as generative adversarial networks on images and transformer-based language models on text, that our bounds exhibit the correct dependence of the estimation error on the sample size $n$ and the support size $k$.

**Related work.** The most widely used metrics for generative models include Inception Score [55], Fréchet Inception Distance [27], and Kernel Inception Distance [4]. The former two are extended to conditional generative models in [2]. They summarize the performance by a single value and thus cannot distinguish different failure cases, i.e., low quality and low diversity. Motivated by this limitation, Sajjadi et al. [54] propose a metric to evaluate the quality of generative models using two separate components: precision and recall. This formulation is extended in [59] to arbitrary probability measures using a density ratio estimator, while alternative definitions based on non-parametric representations of the manifolds of the data were proposed in [37]. These notions are generalized by the divergence frontier framework of Djolonga et al. [18]. Pillutla et al. [49] propose MAUVE, an area-under-the-curve summary based on divergence frontiers for neural text generation. They find that MAUVE correlates well with human judgements on how close the machine generated text and the human text are.

Another line of related work is the estimation of functionals of discrete distributions; see [62] for an overview. In particular, estimation of KL divergences has been studied by [8, 67, 7, 24] in both fixed and large alphabet regimes. These results focus on the expected $\mathbf{L}_1$ and $\mathbf{L}_2$ risks and require additional assumptions on the two distributions such as boundedness of density ratio which is not needed in our results. Recently, Sreekumar et al. [60] investigated a modern way to estimate $f$-divergences using neural networks. On the practical side, there is a new line of successful work that uses deep neural networks to find data-dependent quantizations for the purpose of estimating information theoretic quantities from samples [53, 23].

**Notation.** Let $\mathcal{P}(\mathcal{X})$ be the space of probability distributions on some measurable space $\mathcal{X}$. For any $P, Q \in \mathcal{P}(\mathcal{X})$, let $\mathrm{KL}(P\|Q)$ be the Kullback-Leibler (KL) divergence between $P$ and $Q$. For $\lambda \in (0, 1)$, we define the *interpolated KL divergence* as $\mathrm{KL}_\lambda(P\|Q) := \mathrm{KL}(P\|\lambda P + (1 - \lambda)Q)$. For a partition $\mathcal{S} := \{S_1, \ldots, S_k\}$ of $\mathcal{X}$, we define $P_\mathcal{S}$ the quantized version of $P$ so that $P_\mathcal{S} \in \mathcal{P}(\mathcal{S})$ with $P_\mathcal{S}(S_i) = P(S_i)$ for any $i \in [k] := \{1, \ldots, k\}$.

## 2 Divergence frontiers

Divergence frontiers compare two distributions $P$ and $Q$ using a frontier of statistical divergences. Each point on the frontier compares the individual distributions against a mixture of the two. By sweeping through mixtures, the curve interpolates between measurements of two types of costs. Fig. 1 illustrates divergence frontiers, which we formally introduce below.

**Evaluating generative models via divergence frontiers.** Consider a generative model $Q \in \mathcal{P}(\mathcal{X})$ which attempts to model the target distribution $P \in \mathcal{P}(\mathcal{X})$. It has been argued in [54, 37] that one must consider two types of costs to evaluate $Q$ with respect to $P$: (a) a type I cost (loss in precision), which is the mass of $Q$ that has low or zero probability mass under $P$, and (b) a type II cost (loss in recall), which is the mass of $P$ that $Q$ does not adequately capture.

Suppose $P$ and $Q$ are uniform distributions on their supports, and $R$ is uniform on the union of their supports. Then, the type I cost is the mass of $\mathrm{Supp}(Q) \setminus \mathrm{Supp}(P)$, or equivalently, the mass of $\mathrm{Supp}(R) \setminus \mathrm{Supp}(P)$. We measure this using the surrogate $\mathrm{KL}(Q\|R)$, which is large if there exists $a$ such that $Q(a)$ is large but $R(a)$ is small. Likewise, the type II cost is measured by $\mathrm{KL}(P\|R)$. When $P$ and $Q$ are not constrained to be uniform, it is not clear what the measure $R$ should be. Djolonga et al. [18] propose to vary $R$ over all possible probability measures and consider the Pareto frontier of the multi-objective optimization $\min_R \big(\mathrm{KL}(P\|R), \mathrm{KL}(Q\|R)\big)$. This leads to a curve called the *divergence frontier*, and is reminiscent of the precision-recall curve in binary classification. See [15, 13, 14, 19] and references therein on trade-off curves in machine learning.

Formally, it can be shown that the divergence frontier $\mathcal{F}(P,Q)$ of probability measures $P$ and $Q$ is carved out by mixtures $R_\lambda = \lambda P + (1-\lambda)Q$ for $\lambda \in (0,1)$ (cf. Fig. 1). It admits the closed-form

$$\mathcal{F}(P,Q) = \Big\{ \big(\mathrm{KL}(P\|R_\lambda), \, \mathrm{KL}(Q\|R_\lambda)\big) \, : \, \lambda \in (0,1) \Big\}.$$

**Practical computation of divergence frontiers.** In practical applications, $P$ is a complex, high-dimensional distribution which could either be discrete, as in natural language processing, or continuous, as in computer vision. Likewise, $Q$ is often a deep generative model such as GPT-3 for text and GANs for images. It is infeasible to compute the divergence frontier $\mathcal{F}(P,Q)$ directly because we only have samples from $P$ and the integrals or sums over $Q$ are intractable.

Therefore, the recipe used by practitioners [54, 18, 49] has been to (a) jointly quantize $P$ and $Q$ over a partition $\mathcal{S} = \{S_t\}_{t=1}^k$ of $\mathcal{X}$ to obtain discrete distributions $P_\mathcal{S} = (P(S_t))_{t=1}^k$ and $Q_\mathcal{S} = (Q(S_t))_{t=1}^k$, (b) estimate the quantized distributions from samples to get $\hat{P}_\mathcal{S}$ and $\hat{Q}_\mathcal{S}$, and (c) compute $\mathcal{F}(\hat{P}_\mathcal{S}, \hat{Q}_\mathcal{S})$. In practice, the best quantization schemes are data-dependent transformations such as $k$-means clustering or lattice-type quantization of dense representations of images or text [53].

**Statistical summary of divergence frontiers.** In the minimax theory of hypothesis testing, where the goal is also to study two types of errors (yet different from the ones considered here), it is common to theoretically analyze their linear combination; see, e.g., [30, Sec. 1.2] and [9, Thm. 7]. In the same spirit, we consider a linear combination of the two costs, quantified by the KL divergences,

$$\mathcal{L}_\lambda(P,Q) := \lambda \, \mathrm{KL}(P\|R_\lambda) + (1-\lambda)\mathrm{KL}(Q\|R_\lambda). \tag{1}$$

Note that $R_\lambda$ is exactly the minimizer of the linearized objective $\lambda\mathrm{KL}(P\|R) + (1-\lambda)\mathrm{KL}(Q\|R)$ according to [18, Props. 1 and 2]. $\mathcal{L}_\lambda$ is also known as the $\lambda$-skew Jensen-Shannon Divergence [46].

The linearized cost $\mathcal{L}_\lambda$ depends on the choice of $\lambda$. To remove this dependency, we define a novel integral summary, called the *frontier integral* $\mathrm{FI}(P,Q)$ of two distributions $P$ and $Q$ as

$$\mathrm{FI}(P,Q) := 2 \int_0^1 \mathcal{L}_\lambda(P,Q) \, \mathrm{d}\lambda \, . \tag{2}$$

We can interpret the frontier integral as the average linearized cost over $\lambda \in (0,1)$. While the length of the divergence frontier can be unbounded as shown in Appx. H, the frontier integral is always bounded in $[0,1]$. Moreover, it is a symmetric divergence with $\mathrm{FI}(P,Q) = 0$ iff $P = Q$ (Appx. B). In practice, it can be estimated using the same recipe as the divergence frontier.

**Error decomposition.** In Sec. 3, we decompose the error in estimating the frontier integral into two components: the statistical error of estimating the quantized distribution and the quantization

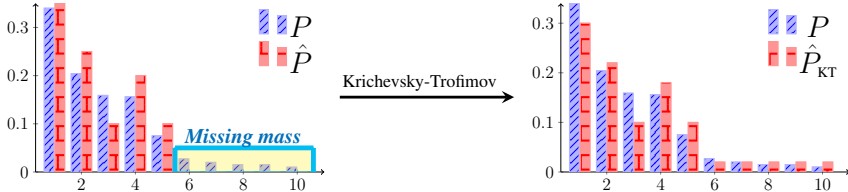

Figure 2: The empirical estimator with missing mass and the Krichevsky-Trofimov estimator.

error. Our goal is to derive the rate of convergence for the overall estimation error. To control the statistical error, we use a different treatment for the masses that appear in the sample and the ones that never appear (i.e., the missing mass). We obtain a high probability bound as well as a bound for its expectation, leading to upper bounds for its sample complexity and rate of convergence. These results carry over to the divergence frontiers as well. As for the quantization error, we construct a distribution-dependent quantization scheme whose error is at most $O(k^{-1})$, where $k$ is the quantization level. A combination of these two bounds sheds light on the optimal choice of the quantization level. In Sec. 5, we verify empirically the tightness of the rates on synthetic and real data.

## 3 Main results

In this section, we summarize our main theoretical results. The results hold for both the linearized cost $\mathcal{L}_\lambda$ and the frontier integral FI, we focus on FI here due to space constraints. For $P, Q \in \mathcal{P}(\mathcal{X})$, let $\{X_i\}_{i=1}^n$ and $\{Y_i\}_{i=1}^n$ be i.i.d. samples from $P$ and $Q$, respectively, and denote by $\hat{P}_n$ and $\hat{Q}_n$ the respective empirical measures of $P$ and $Q$. The two samples are assumed to have the same size $n$ for simplicity. We denote by $C$ an absolute constant which can vary from line to line. The precise statements and proofs can be found in the Appendix.

**Sample complexity for the frontier integral.** We are interested in deriving a non-asymptotic bound for the absolute error of the empirical estimator, i.e., $\left|\mathrm{FI}(\hat{P}_n, \hat{Q}_n) - \mathrm{FI}(P, Q)\right|$. When both $P$ and $Q$ are supported on a finite alphabet with $k$ items, a natural strategy is to exploit the smoothness properties of FI, giving a naïve upper bound $O(L\sqrt{k/n})$ on the absolute error, where $L = \log 1/p_*$ with $p_* = \min_{a \in \mathrm{Supp}(P)} P(a)$ reflecting the smoothness of FI. The dependency on $p_*$ requires $P$ to have a finite support and a short tail. However, in many real-world applications, the distributions can either be supported on a countable set or have long tails [11, 64]. By considering the *missing mass* in the sample, we are able to obtain a high probability bound that is independent of $p_*$.

**Theorem 1.** *Assume that $P$ and $Q$ are discrete and let $k = \max\{|\mathrm{Supp}(P)|, |\mathrm{Supp}(Q)|\} \in \mathbb{N} \cup \{\infty\}$. For any $\delta \in (0,1)$, it holds that, with probability at least $1 - \delta$,*

$$\left|\mathrm{FI}(\hat{P}_n, \hat{Q}_n) - \mathrm{FI}(P, Q)\right| \leq C \left[\left(\sqrt{\frac{\log 1/\delta}{n}} + \alpha_n(P) + \alpha_n(Q)\right) \log n + \beta_n(P) + \beta_n(Q)\right],$$
(3)

*where $\alpha_n(P) = \sum_{a \in \mathcal{X}} \sqrt{n^{-1} P(a)}$ and $\beta_n(P) = \mathbb{E}\left[\sum_{a: \hat{P}_n(a)=0} P(a) \max\left\{1, \log\left(1/P(a)\right)\right\}\right]$. Furthermore, if the support size $k < \infty$, then $\alpha_n(P) \leq \sqrt{k/n}$ and $\beta_n(P) \leq k \log n / n$. In particular, with probability at least $1 - \delta$,*

$$\left|\mathrm{FI}(\hat{P}_n, \hat{Q}_n) - \mathrm{FI}(P, Q)\right| \leq C \left[\sqrt{\frac{\log 1/\delta}{n}} + \sqrt{\frac{k}{n}} + \frac{k}{n}\right] \log n.$$
(4)

Before we discuss the bounds in Thm. 1, let us introduce the missing mass problem. This problem was first studied by Good and Turing [21], where the eponymous Good-Turing estimator was proposed to estimate the probability that a new observation drawn from a fixed distribution has never appeared before, in other words, is missing in the current sample; see Fig. 2 (left) for an illustration. The Good-Turing estimator has been widely used in language modeling [33, 12, 11] and studied in theory [41, 48, 47]. An inspiring result coming from this line of work is that the missing mass in a sample of size $n$ concentrates around its expectation [40], which itself decays as $O(k/n)$ when the distribution is supported on $k$ items [3].

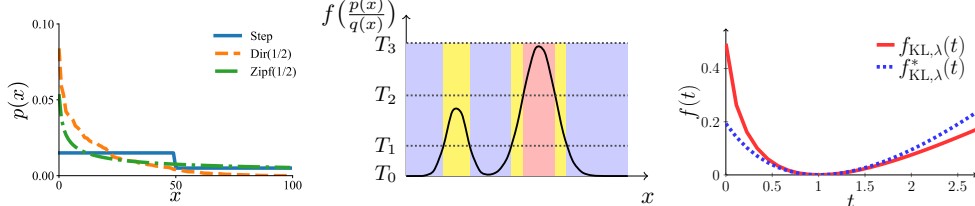

Figure 3: **Left**: Tail decay of three distributions. **Middle**: Oracle quantization into 3 bins: blue, yellow and red. Bin $i$ is given by the set $\{x : f(p(x)/q(x)) \in [T_{i-1}, T_i]\}$. **Right**: The generator and conjugate generator of $\mathrm{KL}_\lambda$ at $\lambda = 1/2$.

There are several merits to Thm. 1. First, (3) holds for any distributions with a countable support. Second, it does not depend on $p_*$ and is adapted to the tail behavior of $P$ and $Q$. For instance, if $P$ is defined as $P(a) \propto a^{-2}$ for $a \in [k]$, then $\alpha_n(P) \propto (\log k)/\sqrt{n}$, which is much better than $\sqrt{k/n}$ in (4) in terms of the dependency on $k$. This phenomenon is also demonstrated empirically in Sec. 5. Third, it captures a parametric rate of convergence, i.e., $O(n^{-1/2})$, up to a logarithmic factor. In fact, this rate is not improvable in a related problem of estimating $\mathrm{KL}(P\|Q)$, even with the assumption that $P/Q$ is bounded [7]. The bound in (4) is a distribution-free bound, assuming $k$ is finite. Note that it also gives an upper bound on the sample complexity by setting the right hand side of (4) to be $\epsilon$ and solving for $n$, this is roughly $O((\sqrt{\log 1/\delta} + \sqrt{k})^2/\epsilon^2)$.

The proof of Thm. 1 relies on two new results: (a) a concentration bound around $\mathbb{E}[\mathrm{FI}(\hat{P}_n, \hat{Q}_n)]$, which can be obtained by McDiarmid's inequality, and (b) an upper bound for the statistical error, i.e., $\mathbb{E}\left|\mathrm{FI}(\hat{P}_n, \hat{Q}_n) - \mathrm{FI}(P, Q)\right|$, which is upper bounded by

$$O\left([\alpha_n(P) + \alpha_n(Q)] \log n + \beta_n(P) + \beta_n(Q)\right) \leq O\left(\left(\sqrt{k/n} + k/n\right) \log n\right). \tag{5}$$

The concentration bound gives the term $\sqrt{n^{-1} \log 1/\delta}$. The statistical error bound is achieved by splitting the masses of $P$ and $Q$ into two parts: one that appears in the sample and one that never appears. The first part can be controlled by a Lipschitz-like property of the frontier integral, leading to the term $\alpha_n(P) + \alpha_n(Q)$, and the second part, $\beta_n(P) + \beta_n(Q)$, falls into the missing mass framework. In addition, the rate $k/n$ for $\beta_n$ shown here matches the rate for the missing mass.

**Statistical consistency of the divergence frontiers.** While Thm. 1 establishes the consistency of the frontier integral, it is also of great interest to know whether the divergence frontier itself can be consistently estimated. In fact, similar bounds hold for the worst-case error of $\mathcal{F}(\hat{P}_n, \hat{Q}_n)$.

**Corollary 2.** *Under the same assumptions as in Thm. 1, the bounds in (3) and (4) hold for*

$$\sup_{\lambda \in [\lambda_0, 1-\lambda_0]} \left\|\left(\mathrm{KL}(\hat{P}_n\|\hat{R}_\lambda), \mathrm{KL}(\hat{Q}_n\|\hat{R}_\lambda)\right) - \left(\mathrm{KL}(P\|R_\lambda), \mathrm{KL}(Q\|R_\lambda)\right)\right\|_1,$$

*where $\hat{R}_\lambda := \lambda\hat{P}_n + (1-\lambda)\hat{Q}_n$, with $C$ replaced by $C/\lambda_0$ for any $\lambda_0 \in (0, 1)$. In particular, if $\lambda_0$ is chosen as $\lambda_n = o(1)$ and $\lambda_n = \omega(\sqrt{k/n}\log n)$, then the expected worst-case error above converges to zero at rate $O(\lambda_n^{-1}\sqrt{k/n}\log n)$.*

The truncation in Cor. 2 is necessary without imposing additional assumptions, since $\mathrm{KL}(P\|R_\lambda)$ is close to $\mathrm{KL}(P\|Q)$ for small $\lambda$ and it is known that the minimax quadratic risk of estimating the KL divergence over all distributions with $k$ bins is always infinity [7].

**Upper bound for the quantization error.** Recall from Sec. 2 that computing the divergence frontiers in practice usually involves a quantization step. Since every quantization will inherently introduce a positive bias in the estimation procedure, it is desirable to control the error, which we call the quantization error, induced by this step. We show that there exists a quantization scheme with error proportional to the inverse of its level. We implement this scheme and empirically verify this rate in Appx. G; certain regimes appear to show even faster convergence.

Let $\mathcal{X}$ be an arbitrary measurable space and $\mathcal{S}$ be a partition of $\mathcal{X}$. The quantization error of $\mathcal{S}$ is the difference $|\mathrm{FI}(P_\mathcal{S}, Q_\mathcal{S}) - \mathrm{FI}(P, Q)|$. It can be shown that there exists a distribution-dependent partition $\mathcal{S}_k$ with level $|\mathcal{S}_k| = k$ whose quantization error is no larger than the inverse of its level, i.e.,

$$|\mathrm{FI}(P, Q) - \mathrm{FI}(P_{\mathcal{S}_k}, Q_{\mathcal{S}_k})| \leq Ck^{-1}. \tag{6}$$

The key idea behind the construction of this partition is visualized in Fig. 3 (middle). Combining this bound with the bounds in (5) leads to the following bound for the total estimation error.

**Theorem 3.** *Assume that $\mathcal{S}_k$ is a partition of $\mathcal{X}$ such that $|\mathcal{S}_k| = k \geq 2$. Then the total error $\mathbb{E}\left|\mathrm{FI}(\hat{P}_{\mathcal{S}_k,n}, \hat{Q}_{\mathcal{S}_k,n}) - \mathrm{FI}(P,Q)\right|$ is upper bounded by*

$$C\left[\,(\alpha_n(P) + \alpha_n(Q))\log n + \beta_n(P) + \beta_n(Q) + |\mathrm{FI}(P,Q) - \mathrm{FI}(P_{\mathcal{S}_k}, Q_{\mathcal{S}_k})|\,\right]. \tag{7}$$

*Moreover, if the quantization error satisfies the bound in (6), we have*

$$\mathbb{E}\left|\mathrm{FI}(\hat{P}_{\mathcal{S}_k,n}, \hat{Q}_{\mathcal{S}_k,n}) - \mathrm{FI}(P,Q)\right| \leq C\left[\left(\sqrt{k/n} + k/n\right)\log n + 1/k\right]. \tag{8}$$

Based on the bound in (8), a good choice of $k$ is $\Theta(n^{1/3})$ which balances between the two types of errors. We illustrate in Sec. 5 that this choice works well in practice. This balancing is enabled by the existence of a good quantizer with a distribution-free bound in (6). In practice, this suggests a data-dependent quantizer using nonparametric density estimators. However, directions such as kernel density estimation [42, 26, 28] and nearest-neighbor methods [1] have not met empirical success, as they suffer from the curse of dimensionality common in nonparametric estimation. In particular, [63, 57, 58] propose quantized divergence estimators but only prove asymptotic consistency, and little progress has been made since then. On the other hand, modern data-dependent quantization techniques based on deep neural networks can successfully estimate properties of the density from high dimensional data [53, 23]. Theoretical results for those techniques could complement our analysis. We leverage these powerful methods to scale our approach on real data in Sec. 5.

## 4 Towards better estimators and interpolated $f$-divergences

**Smoothed distribution estimators.** When the support size $k$ is large, the statistical performance of the empirical estimator considered in the previous section can be improved. To overcome this challenge, practitioners often use more sophisticated distribution estimators such as the Good-Turing estimator [21, 47] and add-constant estimators [35, 5]. We focus on the add-constant estimator defined below and state here its estimation error when it is applied to estimate the frontier integral from data. Again, this result also holds for the linearized cost $\mathcal{L}_\lambda$. We investigate and compare the performance of various distribution estimators in Sec. 5.

For notational simplicity, we assume that $P$ and $Q$ are supported on a common finite alphabet with size $k < \infty$. Note that this is true for the quantized distributions $P_{\mathcal{S}}$ and $Q_{\mathcal{S}}$. For any constant $b > 0$, the add-constant estimator of $P$ is defined as $\hat{P}_{n,b}(a) = (N_a + b)/(n + kb)$ for each $a \in \mathrm{Supp}(P)$, where $N_a = |\{i : X_i = a\}|$ is the number of times $a$ appears in the sample.

Thanks to the smoothing, there is no mass missing in the add-constant estimator. This effect is illustrated for the Krichevsky-Trofimov (add-1/2) estimator in Fig. 2. As a result, we can directly utilize the smoothness properties of the frontier integral to get the following bound.

**Proposition 4.** *Under the same assumptions as in Thm. 3, we have*

$$\mathbb{E}\left|\mathrm{FI}(\hat{P}_{\mathcal{S}_k,n,b}, \hat{Q}_{\mathcal{S}_k,n,b}) - \mathrm{FI}(P,Q)\right|$$
$$\leq C\left[\left(\frac{n(\alpha_n(P) + \alpha_n(Q))}{n + bk} + \gamma_{n,k}(P) + \gamma_{n,k}(Q)\right)\log\left(n/b + k\right) + \frac{1}{k}\right], \tag{9}$$

*where $\gamma_{n,k}(P) = (n + bk)^{-1}bk\sum_{a\in\mathcal{X}}|P(a) - 1/k|$. It can be further upper bounded by $\frac{\sqrt{nk}+bk}{n+bk}\log\left(n/b + k\right) + \frac{1}{k}$ up to a multiplicative constant.*

Let us compare the bounds in Prop. 4 with the ones in Thm. 3. For the distribution-dependent bound, the term $\alpha_n(P)\log n$ in (7) is improved by a factor $n/(n + bk)$ in (9). The missing mass term $\beta_n(P)$ is replaced by the total variation distance between $P$ and the uniform distribution on $[k]$ with a factor $bk/(n + bk)$. The improvements in both two terms are most significant when $k/n$ is large. As for the distribution-free bound, when $k/n$ is small, the bound in Prop. 4 scales the same as the one in (8); when $k/n$ is large (i.e., bounded away from 0 or diverging), it scales as $O(\log n + \log(k/n) + k^{-1})$ while the one in (8) scales as $O(k\log n/n + k^{-1})$. Given the improvement, it would be an interesting venue for future work to consider adaptive estimators in the spirit of [20].

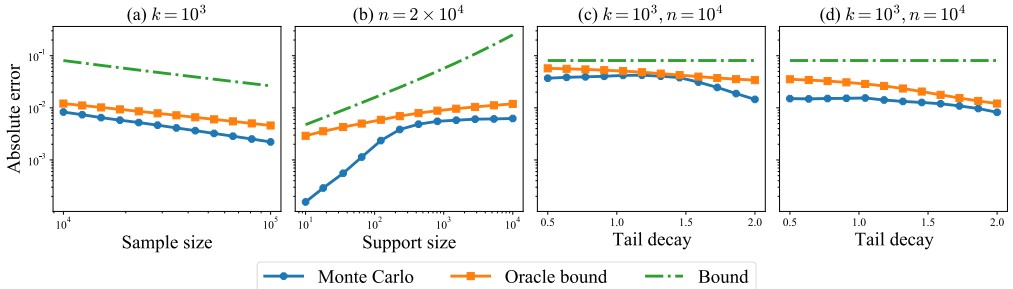

Figure 4: Statistical error of the estimated frontier integral on synthetic data. **(a)**: Zipf(2) and Zipf(2) with $k = 10^3$; **(b)**: Zipf(2) and Zipf(2) with $n = 2 \times 10^4$; **(c)**: Dir($\mathbf{1}$) and Zipf(r) with $k = 10^3$ and $n = 10^4$; **(d)**: Zipf(2) and Zipf(r) with $k = 10^3$ and $n = 10^4$. The bounds are scaled by 100.

**Generalization to $f$-divergences.** Estimation of the $\chi^2$ divergence is useful for variational inference [17] and GAN training [39, 61]. More generally, estimating $f$-divergences from samples is a fundamental problem in machine learning and statistics [45, 29, 10, 52]. We extend our previous results to estimating general $f$-divergences (which satisfy some regularity assumptions) using the same two-step procedure of quantization and estimation of multinomial distributions from samples.

We start by reviewing the definition of $f$-divergences. Let $f : (0, \infty) \to \mathbb{R}$ be a nonnegative and convex function with $f(1) = 0$. Let $P, Q \in \mathcal{P}(\mathcal{X})$ be dominated by some measure $\mu \in \mathcal{P}(\mathcal{X})$ with densities $p$ and $q$, respectively. The $f$-divergence generated by $f$ is defined as

$$D_f(P\|Q) = \int_{\mathcal{X}} q(x) f\left(\frac{p(x)}{q(x)}\right) \mathrm{d}\mu(x),$$

with the convention that $f(0) = f(0^+)$ and $0f(p/0) = pf^*(0)$, where $f^*(0) = f^*(0^+) \in [0, \infty]$ for $f^*(t) = tf(1/t)$. We call $f^*$ the conjugate generator to $f$. An illustration of the generator to $\mathrm{KL}_{1/2}$ can be found in Fig. 3 (right). Note that the conjugacy here is unrelated to the convex conjugacy but is based on the *perspective transform*. The function $f^*$ also generates an $f$-divergence, which is referred to as the *conjugate divergence* to $D_f$ since $D_{f^*}(P\|Q) = D_f(Q\|P)$.

The quantization error bound (6) holds for all $f$-divergences which are *bounded*, i.e., $f(0) + f^*(0) < \infty$. The high probability bounds in Thm. 1 also hold for $f$-divergences, under some regularity assumptions: (a) $|f'(t)| \propto \log t^{-1}$ and $|(f^*)'(t)| \propto \log t^{-1}$ for small $t$, which guarantees that $f$ is approximately Lipschitz and cannot vary too fast; (b) $tf''(t)$ and $t(f^*)''(t)$ are bounded, which is a technical assumption that helps control the variation of $f$ around zero. The interpolated $\chi^2$ divergence, defined analogously as the interpolated KL divergence, satisfies these conditions.

In the Appendix, we prove all the results for general $f$-divergences and show that both the frontier integral and the linearized cost are $f$-divergences satisfying the regularity conditions, recovering Thm. 1 and Thm. 3 as special cases.

## 5 Experiments

We investigate the empirical behavior of the divergence frontier and the frontier integral on both synthetic and real data. Our main findings are: (a) the statistical error bound approximately reveals the rate of convergence of the empirical estimator; (b) the smoothed distribution estimators improve the estimation accuracy; (c) the quantization level suggested by the theory works well empirically. The results for the divergence frontier and the frontier integral are almost identical. We focus on the latter here due to space constraints. In all the plots, we visualize the average absolute error computed from 100 repetitions with shaded region denoting one standard deviation around the mean. More details and additional results, including the ones for the divergence frontier, are deferred to Appx. G. The code to reproduce the experiments is available online[1].

### 5.1 Experimental setup

We work with synthetic data in the case when $k = |\mathcal{X}| < \infty$ as well as real image and text data.

---

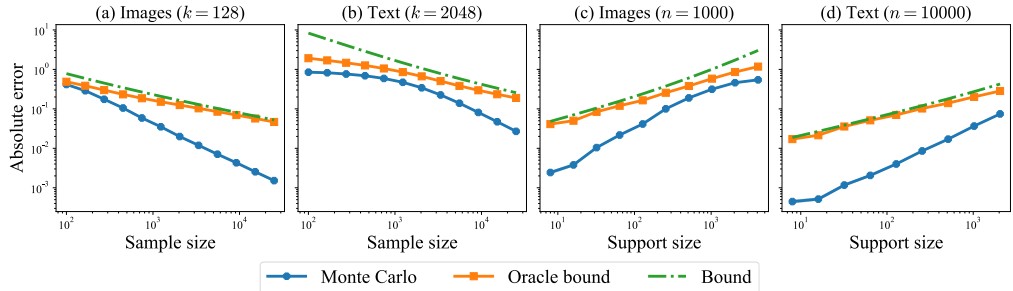

Figure 5: Statistical error of the estimated frontier integral on real data. **(a)**: Image data (CIFAR-10) with $k = 128$; **(b)**: Text data (WikiText-103) with $k = 2048$; **(c)**: Image data (CIFAR-10) with $n = 1000$; **(d)**: Text data (WikiText-103) with $n = 10000$. The bounds are scaled by 30.

**Synthetic Data.** Following the experimental settings in [47], we consider three types of distributions: (a) the Zipf$(r)$ distribution with $r \in \{0, 1, 2\}$ where $P(i) \propto i^{-r}$. Note that Zipf$(r)$ is regularly varying with index $-r$; see, e.g., [56, Appx. B]. (b) the Step distribution where $P(i) = 1/2$ for the first half bins and $P(i) = 3/2$ for the second half bins. (c) the Dirichlet distribution $\mathrm{Dir}(\alpha)$ with $\alpha \in \{\mathbf{1}/2, \mathbf{1}\}$; see Fig. 1 (left) for an illustration. In total, there are 6 different distributions, giving 21 different pairs of $(P, Q)$. For each pair $(P, Q)$, we generate i.i.d. samples of size $n$ from each of them, and estimate the divergence frontier as well as the frontier integral from these samples.

**Real Data.** We consider two domains: images and text. For the image domain, we train a Style-GAN2 [32] on the CIFAR-10 dataset [36] using the publicly available code[2] with default hyperparameters. To evaluate the divergence frontiers, we use the test set of 10k images as the target distribution $P$ and we sample 10k images from the generative model as the model distribution $Q$. For the text domain, we fine-tune a pretrained GPT-2 [50] model with 124M parameters (i.e., GPT-2 small) on the Wikitext-103 dataset [43]. We use the open-source HuggingFace Transformers library [65] for training, and generate 10k 500-token completions using top-$p$ sampling and 100-token prefixes.

We take the following steps to compute the frontier integral. First, we represent each image/text by its features [27, 54, 37]. Second, we learn a low-dimensional feature embedding which maintains the neighborhood structure of the data while encouraging the features to be uniformly distributed on the unit sphere [53]. Third, we quantize these embeddings on a uniform lattice with $k$ bins. For each support size $k$, this gives us quantized distributions $P_{\mathcal{S}_k}$ and $Q_{\mathcal{S}_k}$. Finally, we sample $n$ i.i.d. observations from each of these distributions and consider the empirical distributions $\hat{P}_{\mathcal{S}_k, n}$ and $\hat{Q}_{\mathcal{S}_k, n}$ as well as the smoothed distribution estimators computed from these samples.

**Performance Metric.** We are interested in the estimation of the frontier integral $\mathrm{FI}(P, Q)$ using estimators $\mathrm{FI}(\hat{P}_n, \hat{Q}_n)$ for the empirical estimator as well as the smoothed distribution estimator. We measure the quality of estimation using the absolute error, which is defined as $|\mathrm{FI}(\hat{P}_n, \hat{Q}_n) - \mathrm{FI}(P, Q)|$. For the real data, we measure the error of estimating $\mathrm{FI}(P_{\mathcal{S}_k}, Q_{\mathcal{S}_k})$ by $\mathrm{FI}(\hat{P}_{\mathcal{S}_k, n}, \hat{Q}_{\mathcal{S}_k, n})$.

### 5.2 Tightness of the Statistical Bound

We investigate the tightness of the statistical error bound of Thm. 1 with respect to the sample size $n$ and the support size $k$, in order to verify the validity of the theory in practically relevant settings.

We estimate the expected absolute error $\mathbb{E}|\mathrm{FI}(\hat{P}_n, \hat{Q}_n) - \mathrm{FI}(P, Q)|$ from a Monte Carlo estimate using 100 random trials. We compare it with the following bounds in[3] Thm. 1:

(a) **Bound**: the distribution independent bound $(\sqrt{k/n} + k/n) \log n$.

(b) **Oracle Bound**: the distribution dependent bound $(\alpha_n(P) + \alpha_n(Q)) \log n + \beta_n(P) + \beta_n(Q)$. We assume that the quantities $\alpha_n$ and $\beta_n$ defined in Thm. 1 are known.

---

[2]https://github.com/NVlabs/stylegan2-ada-pytorch.
[3]Specifically, we use the expected bound of Prop. 16 (Appx. D), from which Thm. 1 is derived.

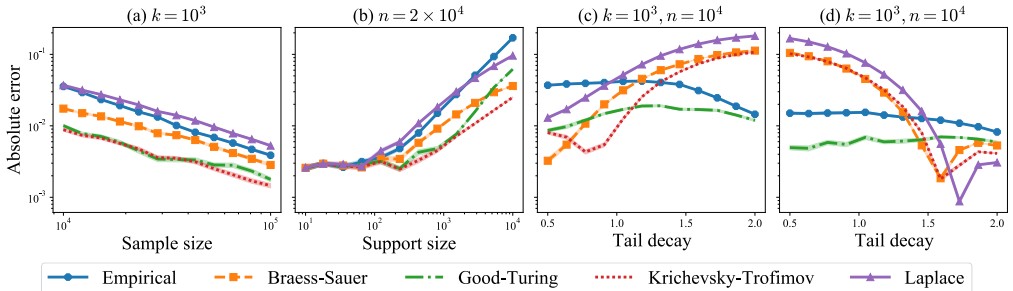

Figure 6: Statistical error with smoothed distribution estimators on synthetic data. **(a)**: $\mathrm{Zipf}(0)$ and $\mathrm{Dir}(\mathbf{1}/2)$ with $k = 10^3$; **(b)**: $\mathrm{Zipf}(0)$ and $\mathrm{Dir}(\mathbf{1}/2)$ with $n = 2 \times 10^4$; **(c)**: $\mathrm{Dir}(\mathbf{1})$ and $\mathrm{Zipf}(r)$ with $k = 10^3$ and $n = 10^4$; **(d)**: $\mathrm{Zipf}(2)$ and $\mathrm{Zipf}(r)$ with $k = 10^3$ and $n = 10^4$.

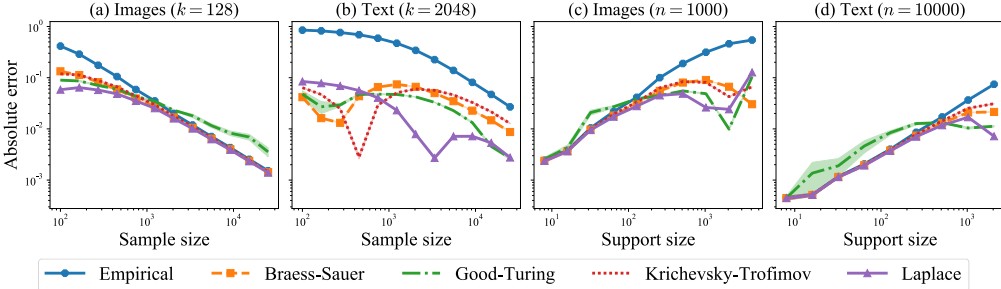

Figure 7: Statistical error with smoothed distribution estimators on real data. **(a)**: Image data (CIFAR-10) with $k = 128$; **(b)**: Text data (WikiText-103) with $k = 2048$; **(c)**: Image data (CIFAR-10) with $n = 1000$; **(d)**: Text data (WikiText-103) with $n = 10000$. The bounds are scaled by $15$.

We fix $k$, plot each of these quantities in a log-log plot with varying $n$ and compare their *slopes*.[4] We then repeat the experiment with $n$ fixed and $k$ varying. We often scale the bounds by a constant for easier visual comparison of the slopes; this only changes the intercept and leaves the slope unchanged.

**Thm. 1 is tight for synthetic data.** Fig. 4 gives the Monte Carlo estimate and the bounds of the statistical error for various synthetic data distributions. In Fig. 4(a), we observe that the bound has approximately the same slope as the Monte Carlo estimate, while the oracle bound has a slightly worse slope. In Fig. 4(b), we observe that the oracle bound captures the correct rate for $k > 300$, while the distribution-independent bound captures the correct rate at small $k$. For the right two plots, both bounds capture the right rate over a wide range of tail decay. The oracle bound is tighter for fast decay, where the distribution-independent bounds on $\alpha_n(Q)$ and $\beta_n(Q)$ can be very pessimistic.

**Thm. 1 is somewhat tight for real data.** Fig. 5 contains the analogous plot for real data, where the observations are similar. In Fig. 5(b), we see that the oracle bound captures the right rate for small sample sizes where $k/n > 1$. However, for large $n$, the distribution-independent bound is better at matching the slope of the Monte Carlo estimate. The same is true for Fig. 5(c), where the oracle bound is better for large $k$. For parts (a) and (d), however, both bounds do not capture the right slope of the Monte Carlo estimate; Thm. 1 is not a tight upper bound in this case.

### 5.3 Effect of Smoothed Distribution Estimators

We now show that smoothed estimators can lead to improved estimation over the naïve empirical estimator and thus improved sample complexity as shown in Prop. 4. This is practically significant in the context of generative models, since one can have an equally good estimate of the divergence frontier with fewer samples using smoothed estimators [54, 18].

Concretely, we compare the Monte Carlo estimates of the absolute error $\mathbb{E}|\mathrm{FI}(\hat{P}_n, \hat{Q}_n) - \mathrm{FI}(P, Q)|$ for the plug-in estimate (denoted "Empirical") with the corresponding estimates for smoothed estimators.

---

[4]A log-log plot of the function $f(x) = cx^\gamma$ is a straight line with slope $\gamma$, which thus captures the *degree*.

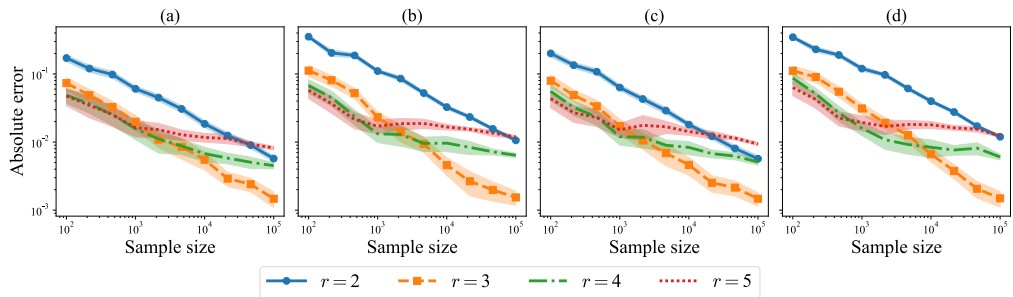

Figure 8: Total error with quantization level $k \propto n^{1/r}$ on 2-dimensional continuous data. **(a)**: $\mathcal{N}(0, I_2)$ and $\mathcal{N}(1, I_2)$; **(b)**: $\mathcal{N}(0, I_2)$ and $\mathcal{N}(0, 5I_2)$; **(c)**: $t_4(0, I_2)$ and $t_4(1, I_2)$ (multivariate t-distribution with 4 degrees of freedom); **(d)**: $t_4(0, I_2)$ and $t_4(0, 5I_2)$.

We consider 4 smoothed estimators as in [47]: the (modified) *Good-Turing* estimator, as well as three add-constant estimators: the *Laplace*, *Krichevsky-Trofimov* and *Braess-Sauer* estimators.

**Smoothed estimators are more efficient than the empirical estimator.** We compare the smoothed estimators to the empirical one in Fig. 6 on synthetic data and Fig. 7 on real data. In general, the smoothed distribution estimators reduces the absolute error. For parts (a) and (b) of Fig. 6, the Good-Turing and the Krichevsky-Trofimov estimators have the best absolute error. For parts (c) and (d), the Good-Turing estimator is adapted to various regimes of tail-decay, outperforming the empirical estimator. The Krichevsky-Trofimov and Braess-Sauer estimators, on the other hand, exhibit small absolute error for particular decay regimes. The results are similar for real data in Fig. 7.

**Practical guidance on choosing a smoothed estimator.** While the smoothed estimators offer a marked improvement when $k/n$ is large (that is, close to 1), the best estimator is problem-dependent. As a rule of thumb, we suggest the Krichevsky-Trofimov estimator which works well in the large $k/n$ regime but is still competitive when $k/n$ is small (i.e., large $n$).

### 5.4 Quantization Error

Next, we study the effect of the quantization level $k$ on the total error. We consider a simple 2-dimensional synthetic setting where the distributions $P, Q$ are either the multivariate normal or $t$-distributions. We use data-driven quantization with $k$-means to obtain a quantization $\mathcal{S}_k$: each component of the partition is the region corresponding to one cluster. Finally, we plot the absolute error $\mathbb{E}|\mathrm{FI}(P, Q) - \mathrm{FI}(\hat{P}_{\mathcal{S}_k, n}, \hat{Q}_{\mathcal{S}_k, n})|$, where the $\mathrm{FI}(P, Q)$ is computed using numerical integration and the expectation is estimated with Monte Carlo simulations.

**The choice $k = \Theta(n^{1/3})$ works the best.** We compare $k = n^{1/r}$ for $r = 2, 3, 4, 5$ in Fig. 8. For small $n$, $r \geq 3$ all perform similarly, but $r = 3$ clearly outperforms other choices for $n \geq 10^4$. While our theory does not directly apply for data-dependent partitioning schemes, the choice $k = \Theta(n^{1/3})$ suggested by Thm. 3 nevertheless works well in practice.

## 6 Conclusion

In this paper, we study the statistical behavior of the divergence frontiers and the proposed integral summary estimated from data. We decompose the estimation error into two components, the statistical error and the quantization error, to conform with the approximation procedure commonly used in practice. We establish non-asymptotic bounds on both of the two errors. Our bounds shed light on the optimal choice of the quantization level $k$—they suggests that the two errors can be balanced at $k = \Theta(n^{1/3})$. We also derive a new concentration inequality for the frontier integral, which provides the sample complexity of achieving a small error in high probability. Finally, we demonstrate both theoretically and empirically that the use of smoothed distribution estimators can improve the estimation accuracy. All the results can be generalized to a large class of interpolation-based $f$-divergences. Provided new theoretical results on modern data-dependent quantization schemes using deep neural networks, it would be an interesting direction for future work to specialize our bounds to such quantization schemes. Extending our results to $\beta$-divergences could also be interesting.

**Acknowledgments.** The authors would like to thank J. Thickstun for fruitful discussions. Part of this work was done while Z. Harchaoui was visiting the Simons Institute for the Theory of Computing. This work was supported by NSF DMS-2134012, NSF CCF-2019844, the CIFAR program "Learning in Machines and Brains", and faculty research awards.

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
