# Appendix

## Table of Contents

# A  $f$-divergence: review and examples

We review the definition of $f$-divergences and give a few examples.

Let $f : (0, \infty) \to \mathbb{R}$ be a convex function with $f(1) = 0$. Let $P, Q \in \mathcal{P}(\mathcal{X})$ be dominated by some measure $\mu \in \mathcal{P}(\mathcal{X})$ with densities $p$ and $q$, respectively. The $f$-divergence generated by $f$ is

$$D_f(P\|Q) = \int_{\mathcal{X}} q(x) f\left(\frac{p(x)}{q(x)}\right) \mathrm{d}\mu(x),$$

with the convention that $f(0) := \lim_{t \to 0^+} f(t)$ and $0f(p/0) = pf^*(0)$, where $f^*(0) = \lim_{x \to 0^+} xf(1/x) \in [0, \infty]$. Hence, $D_f(P\|Q)$ can be rewritten as

$$D_f(P\|Q) = \int_{q>0} q(x) f\left(\frac{p(x)}{q(x)}\right) \mathrm{d}\mu(x) + f^*(0)P[q = 0],$$

with the agreement that the last term is zero if $P[q = 0] = 0$ no matter what value $f^*(0)$ takes (which could be infinity). For any $c \in \mathbb{R}$, it holds that $D_{f_c}(P\|Q) = D_f(P\|Q)$ where $f_c(t) = f(t) + c(t-1)$. Hence, we also assume, w.l.o.g., that $f(t) \geq 0$ for all $t \in (0, \infty)$. To summarize, $f$ is convex and nonnegative with $f(1) = 0$. As a result, $f$ is non-increasing on $(0, 1]$ and non-decreasing on $[1, \infty)$.

The conjugate generator to $f$ is the function $f^* : (0, \infty) \to [0, \infty)$ defined by[5]

$$f^*(t) = tf(1/t),$$

where again we define $f^*(0) = \lim_{t \to 0^+} f^*(t)$. Since $f^*$ can be constructed by the perspective transform of $f$, it is also convex. We can verify that $f^*(1) = 0$ and $f^*(t) \geq 0$ for all $t \in (0, \infty)$, so it defines another divergence $D_{f^*}$. We call this the *conjugate divergence* to $D_f$ since

$$D_{f^*}(P\|Q) = D_f(Q\|P).$$

The divergence $D_f$ is symmetric if and only if $f = f^*$, and we write it as $D_f(P, Q)$ to emphasize the symmetry.

**Example 5.** We illustrate a number of examples.

(a) KL divergence: It is an $f$-divergence generated by $f_{\mathrm{KL}}(t) = t \log t - t + 1$.

(b) Interpolated KL divergence: For $\lambda \in (0, 1)$, the interpolated KL divergence is defined as

$$\mathrm{KL}_\lambda(P\|Q) = \mathrm{KL}(P\|\lambda P + (1 - \lambda)Q),$$

which is a $f$-divergence generated by

$$f_{\mathrm{KL},\lambda}(t) = t \log\left(\frac{t}{\lambda t + 1 - \lambda}\right) - (1 - \lambda)(t - 1).$$

(c) Jensen-Shannon divergence: The Jensen-Shannon Divergence is defined as

$$D_{\mathrm{JS}}(P, Q) = \frac{1}{2}\mathrm{KL}_{1/2}(P\|Q) + \frac{1}{2}\mathrm{KL}_{1/2}(Q\|P).$$

More generally, we have the $\lambda$-skew Jensen-Shannon Divergence [46], which is defined for $\lambda \in (0, 1)$ as $D_{\mathrm{JS},\lambda} = \lambda \mathrm{KL}_\lambda(P\|Q) + (1 - \lambda)\mathrm{KL}_{1-\lambda}(Q\|P)$. This is an $f$-divergence generated by

$$f_{\mathrm{JS},\lambda}(t) = \lambda t \log\left(\frac{t}{\lambda t + 1 - \lambda}\right) + (1 - \lambda) \log\left(\frac{1}{\lambda t + 1 - \lambda}\right).$$

Note that this is the linearized cost defined in (1)

(d) Frontier Integral: From Prop. 6, FI is an $f$-divergence generated by

$$f_{\mathrm{FI}}(t) = \frac{t + 1}{2} - \frac{t}{t - 1} \log t.$$

---

[5]The conjugacy between $f$ and $f^*$ is unrelated to the usual Fenchel or Lagrange duality in convex analysis, but is related to the perspective transform.

(e) Interpolated $\chi^2$ divergence: Similar to the interpolated KL divergence, we can define the interpolated $\chi^2$ divergence $D_{\chi^2,\lambda}$ and the corresponding convex generator $f_{\chi^2,\lambda}$ for $\lambda \in (0,1)$ as

$$D_{\chi^2,\lambda}(P\|Q) = D_{\chi^2}(P\|\lambda P + (1-\lambda)Q), \quad \text{and,} \quad f_{\chi^2,\lambda}(t) = \frac{(t-1)^2}{\lambda t + 1 - \lambda}.$$

The usual Neyman and Pearson $\chi^2$ divergences are respectively obtained in the limits $\lambda \to 1$ and $\lambda \to 0$.

(f) Squared Le Cam distance: The squared Le Cam distance is, up to scaling, a special case of the interpolated $\chi^2$ divergence with $\lambda = 1/2$:

$$D_{\mathrm{LC}}(P,Q) = \frac{1}{4}D_{\chi^2,1/2}(P\|Q).$$

(g) Squared Hellinger Distance: It is an $f$-divergence generated by $f_H(t) = (1-\sqrt{t})^2$.

## B  Properties of the frontier integral

We prove some properties of the frontier integral here.

First, the frontier integral can be computed in closed form as below.

**Proposition 6.** *Let $P$ and $Q$ be dominated by some probability measure $\mu$ with density $p$ and $q$, respectively. Then,*

$$\mathrm{FI}(P,Q) = \int_{\mathcal{X}} \mathbb{1}\{p(x) \neq q(x)\} \left( \frac{p(x)+q(x)}{2} - \frac{p(x)q(x)}{p(x)-q(x)} \log \frac{p(x)}{q(x)} \right) \mathrm{d}\mu(x), \quad (10)$$

*with the convention $0\log 0 = 0$. Moreover, FI is an $f$-divergence generated by the convex function*

$$f_{\mathrm{FI}}(t) = \frac{t+1}{2} - \frac{t}{t-1}\log t,$$

*with the understanding that $f_{\mathrm{FI}}(1) = \lim_{t \to 1} f_{\mathrm{FI}}(t) = 0$.*

*Proof of Prop. 6.* Let $\bar{\lambda} = 1 - \lambda$. By Tonelli's theorem, it holds that $\mathrm{FI}(P,Q) = 2\int_{\mathcal{X}} h(p(x),q(x))\mathrm{d}\mu(x)$, where

$$h(p,q) = \int_0^1 \left( \lambda p \log p + \bar{\lambda} q \log q - (\lambda p + \bar{\lambda} q)\log(\lambda p + \bar{\lambda} q) \right) \mathrm{d}\lambda.$$

When $p = q$, the integrand is 0. If $q = 0$, then the second term inside the integral is 0, while the first term is

$$\int_0^1 \lambda p \log\frac{1}{\lambda}\mathrm{d}\lambda = \frac{p}{4}.$$

Finally, when $p \neq q$ are both non-zero, we evaluate the integral to get,

$$h(p,q) = \frac{p}{2}\log p + \frac{q}{2}\log q - \frac{2p^2\log p - p^2 - 2q^2\log q + q^2}{4(p-q)},$$

and rearranging the expression completes the proof. $\qquad \square$

Next, the frontier integral is symmetric and bounded.

**Proposition 7.** *The frontier integral satisfies the following properties:*

*(a)* $\mathrm{FI}(P,Q) = \mathrm{FI}(Q,P)$.

*(b)* $0 \leq \mathrm{FI}(P,Q) \leq 1$ *with* $\mathrm{FI}(P,Q) = 0$ *if and only if* $P = Q$.

*Proof of Prop. 7.* The first part follows from the closed form expression in Prop. 6. For the second part, we get the upper bound as

$$\mathrm{FI}(P,Q) \leq \int_{\mathcal{X}} \frac{p(x)+q(x)}{2}\mathrm{d}\mu(x) = 1.$$

We have $\mathrm{FI}(P,Q) \geq 0$ with $\mathrm{FI}(P,P) = 0$ since FI is an $f$-divergence. Further, since $f_{\mathrm{FI}}$ is strictly convex at 1, we get that $\mathrm{FI}(P,Q) = 0$ only if $P = Q$. $\qquad \square$

## C    Regularity assumptions

In this section, we state and discuss the regularity assumptions required for the statistical error bounds. Throughout, we assume that $\mathcal{X}$ is a finite set (for instance, on the quantized space). We upper bound the expected error of the empirical $f$-divergences estimated from data.

We use the convention that all higher order derivatives of $f$ and $f^*$ at $0$ are defined as the corresponding limits as $x \to 0^+$ (if they exist). Further, we use the notation

$$\psi(p, q) = qf(p/q) = pf^*(q/p), \tag{11}$$

so that $D_f(P\|Q) = \sum_{a \in \mathcal{X}} \psi(P(a), Q(a))$.

### C.1    Assumptions

We make the following assumptions about the functions $f$ and $f^*$.

**Assumption 8.** The generator $f$ is twice continuously differentiable with $f'(1) = 0$. Moreover,

- **(A1)** We have $C_0 := f(0) < \infty$ and $C_0^* := f^*(0) < \infty$.
- **(A2)** There exist constants $C_1, C_1^* < \infty$ such that for every $x \in (0, 1)$, we have,
$$|f'(t)| \leq C_1 \left(1 \vee \log 1/t\right), \quad \text{and,} \quad |(f^*)'(t)| \leq C_1^* \left(1 \vee \log 1/t\right) .$$
- **(A3)** There exist constants $C_2, C_2^* < \infty$ such that for every $t \in (0, \infty)$, we have,
$$\frac{t}{2} f''(t) \leq C_2, \quad \text{and,} \quad \frac{t}{2}(f^*)''(t) \leq C_2^* .$$

**Remark 9.** *We discuss the asymptotics of the assumptions.*

- *(a) Assumption (A1) ensures boundedness of the $f$-divergence. Indeed, $f(0) = \infty$ leads to $D_f(P\|Q) = \infty$ if there exists an atom $a \in \mathcal{X}$ such that $P(a) = 0$ but $Q(a) \neq 0$. This happens, for instance, with the reverse KL divergence ($f(t) = -\log t + t - 1$). By symmetry, $f^*(0) = \infty$ leads to a case where $D_f(P\|Q) = \infty$ if there exists an atom $a \in \mathcal{X}$ such that $Q(a) = 0$ but $P(a) \neq 0$, as in the (forward) KL divergence.*
- *(b) Since $f'$ is monotonic nondecreasing and $f'(1) = 0$, we have that $f'(0) \leq 0$ (with strict inequality if $f$ is strictly convex at $1$). In fact, $f'(0) = -\infty$ for each of the divergences considered in Example 5. Assumption (A2) requires $f'(t)$ to behave as $\log 1/t$ when $t \to 0$. Likewise for $(f^*)'$.*
- *(c) Likewise, we have that $f''(0) = \infty$ and $f''(\infty) = 0$ for each of the divergence considered in Example 5. However, Assumption (A3) makes assumptions on the rates of these limits. Namely, $f''$ should diverge no faster than $1/t$ as $t \to 0$ and $f''$ should converge to $0$ at least as fast as $1/t^2$ as $t \to \infty$. We can summarize the implied asymptotics of $f''$ as*
$$f''(t) = \begin{cases} \Omega(1/t), & \text{if } t \to 0, \\ O(1/t^2), & \text{if } t \to \infty. \end{cases}$$

### C.2    Examples satisfying the assumptions

We now consider the examples in Example 5. The constants are summarized in Tab. 1.

**KL divergence.** We have

$$f_{\text{KL}}(t) = t \log t - t + 1 \quad \text{and} \quad f^*_{\text{KL}}(t) = -\log t + t - 1 .$$

We have $f(0) = 1$ but $f^*(0) = \infty$. Therefore, the KL divergence does not satisfy our assumptions. Indeed, this is because the KL divergence can be unbounded.

**Interpolated KL Divergence.** Let $\lambda \in (0, 1)$ be a parameter and denote $\bar{\lambda} = 1 - \lambda$. We have

$$f_{\text{KL},\lambda}(t) = t \log\left(\frac{t}{\lambda t + \bar{\lambda}}\right) - \bar{\lambda}(t - 1) \quad \text{and} \quad f^*_{\text{KL},\lambda}(t) = -\log(\bar{\lambda}t + \lambda) + \bar{\lambda}(t - 1) .$$

The corresponding derivatives are

$$f'_{\text{KL},\lambda}(t) = \frac{\bar{\lambda}}{\lambda t + \bar{\lambda}} + \log\left(\frac{t}{\lambda t + \bar{\lambda}}\right) - \bar{\lambda}, \qquad (f^*_{\text{KL},\lambda})'(t) = \bar{\lambda} - \frac{\bar{\lambda}}{\bar{\lambda}t + \lambda},$$

$$f''_{\text{KL},\lambda}(t) = \frac{\bar{\lambda}^2}{t(\lambda t + \bar{\lambda})^2}, \qquad (f^*_{\text{KL},\lambda})''(t) = \frac{\bar{\lambda}^2}{(\bar{\lambda}t + \lambda)^2} .$$

Table 1: Examples of $f$-divergences and whether they satisfy Assumptions (A1)-(A3). Here, $\lambda \in (0,1)$ is a parameter of the interpolated or skew divergences, and we define $\bar\lambda := 1 - \lambda$.

| $f$-divergence | Satisfies Assumptions? | $C_0$ | $C_0^*$ | $C_1$ | $C_1^*$ | $C_2$ | $C_2^*$ |
|---|---|---|---|---|---|---|---|
| KL | No | $1$ | $\infty$ | | | | |
| Interpolated KL | Yes | $\bar\lambda$ | $\log\frac{1}{\lambda} - \bar\lambda$ | $1$ | $\frac{\bar\lambda^2}{\lambda}$ | $\frac{1}{2}$ | $\frac{\bar\lambda}{8\lambda}$ |
| JS | Yes | $\frac{1}{2}\log 2$ | $\frac{1}{2}\log 2$ | $\frac{1}{2}$ | $\frac{1}{2}$ | $\frac{1}{4}$ | $\frac{1}{4}$ |
| Skew JS | Yes | $\bar\lambda\log\frac{1}{\bar\lambda}$ | $\lambda\log\frac{1}{\lambda}$ | $\lambda$ | $\bar\lambda$ | $\frac{\lambda}{2}$ | $\frac{\bar\lambda}{2}$ |
| Frontier integral | Yes | $\frac{1}{2}$ | $\frac{1}{2}$ | $4$ | $4$ | $\frac{1}{2}$ | $\frac{1}{2}$ |
| LeCam | Yes | $\frac{1}{2}$ | $\frac{1}{2}$ | $2$ | $2$ | $\frac{8}{27}$ | $\frac{8}{27}$ |
| Interpolated $\chi^2$ | Yes | $\frac{1}{\lambda}$ | $\frac{1}{\lambda}$ | $\frac{2}{\bar\lambda^2}$ | $\frac{2}{\bar\lambda^2}$ | $\frac{4}{27\lambda\bar\lambda^2}$ | $\frac{4}{27\lambda^2\bar\lambda}$ |
| Hellinger | No | $1$ | $1$ | $\infty$ | $\infty$ | | |

**Proposition 10.** *The interpolated KL divergence generated by $f_{\mathrm{KL},\lambda}$ satisfies Assumption 8 with*

$$
C_0 = 1-\lambda, \quad C_0^* = \log\frac{1}{\lambda} - 1 + \lambda, \quad C_1 = 1, \quad C_1^* = \frac{(1-\lambda)^2}{\lambda}, \quad C_2 = \frac{1}{2}, \quad C_2^* = \frac{1-\lambda}{8\lambda}.
$$

*Proof.* First, $C_0, C_0^*$ can be computed directly. Second, it is clear that

$$
-f_{\mathrm{KL},\lambda}'(t) = \log\frac{1}{t} + \log(\lambda t + \bar\lambda) - \frac{\bar\lambda}{\lambda t + \bar\lambda} + \bar\lambda \le \log\frac{1}{t} + \log 1 - \bar\lambda + \bar\lambda = \log\frac{1}{t}
$$

for all $x \in (0,1)$. Moreover, since $f$ is convex and $f_{\mathrm{KL},\lambda}'(1) = 0$, it holds that $f_{\mathrm{KL},\lambda}'(x) \le 0$ for all $x \in (0,1)$, and thus $C_1 = 1$. Next, we note that $|(f_{\mathrm{KL},\lambda}^*)'(x)| \le \bar\lambda^2/\lambda$ holds uniformly on $(0,1)$ (or equivalently that $f_{\mathrm{KL},\lambda}^*$ is Lipschitz); this gives $C_1^*$. Next, we have

$$
C_2 = \sup_{t>0}\left\{\frac{1}{2}t f_{\mathrm{KL},\lambda}''(t)\right\} \le \frac{1}{2},
$$

since the function inside the sup is monotonic decreasing on $(0,\infty)$. Finally, we have

$$
C_2^* = \sup_{t>0}\left\{\frac{1}{2}t (f_{\mathrm{KL},\lambda}^*)''(t)\right\} = \frac{\bar\lambda}{8\lambda},
$$

since the term inside the sup is maximized at $t = \lambda/\bar\lambda$. $\qquad\square$

**Skew Jensen-Shannon Divergence.** Let $\lambda \in (0,1)$ be a parameter and $\bar\lambda = 1-\lambda$. We have,

$$
f_{\mathrm{JS},\lambda}(t) = \lambda t \log\left(\frac{t}{\lambda t + \bar\lambda}\right) + \bar\lambda\log\left(\frac{1}{\lambda t + \bar\lambda}\right) = f_{\mathrm{JS},1-\lambda}^*(t).
$$

Its derivatives are

$$
f_{\mathrm{JS},\lambda}'(t) = \lambda\log\left(\frac{t}{\lambda t + \bar\lambda}\right) \quad\text{and}\quad f_{\mathrm{JS},\lambda}''(t) = \frac{\lambda\bar\lambda}{t(\lambda t + \bar\lambda)}.
$$

**Proposition 11.** *The $\lambda$-skew JS divergence generated by $f_{\mathrm{JS},\lambda}$ above satisfies Assumption 8 with*

$$
C_0 = (1-\lambda)\log\frac{1}{1-\lambda}, \quad C_0^* = \lambda\log\frac{1}{\lambda}, \quad C_1 = \lambda, \quad C_1^* = 1-\lambda, \quad C_2 = \frac{\lambda}{2}, \quad C_2^* = \frac{1-\lambda}{2}.
$$

*Proof.* For $C_1$, we have

$$
-f_{\mathrm{JS},\lambda}'(t) = \lambda\log\frac{1}{t} + \lambda\log(\lambda t + \bar\lambda) \le \lambda\log\frac{1}{t}
$$

for $x \in (0,1)$. Next, we have

$$C_2 = \frac{\lambda\bar{\lambda}}{2} \sup_{t>0} \frac{1}{\lambda t + \bar{\lambda}} = \frac{\lambda}{2}.$$

$\square$

**Frontier integral.** We have

$$f_{\mathrm{FI}}(t) = \frac{t+1}{2} - \frac{t}{t-1} \log t = f_{\mathrm{FI}}^*(t).$$

Its derivatives are

$$f_{\mathrm{FI}}'(t) = \frac{(1-t)(3-t) + 2\log t}{2(1-t)^2} \quad \text{and} \quad f_{\mathrm{FI}}''(t) = \frac{2t\log t - t^2 + 1}{t(1-t)^3}.$$

**Proposition 12.** *The frontier integral satisfies Assumption 8 with*

$$C_0 = \frac{1}{2} = C_0^*, \quad C_1 = 1 = C_1^*, \quad C_2 = \frac{1}{2} = C_2^*.$$

*Proof.* We get $C_0$ by calculating the limit as $x \to 0$ using L'Hôpital's rule. For $C_2$, we note that the term inside the sup below is decreasing in $x$ to get

$$C_2 = \sup_{t>0} \frac{2t\log t - t^2 + 1}{(1-t)^3} = \frac{1}{2}.$$

By definition,

$$f_{\mathrm{FI}}(t) = 2\int_0^1 f_{\mathrm{JS},\lambda}(t)\mathrm{d}\lambda,$$

so that, by Prop. 11,

$$-f_{\mathrm{FI}}'(t) = -2\int_0^1 f_{\mathrm{JS},\lambda}'(t)\mathrm{d}\lambda \le 2\int_0^1 \lambda \log\frac{1}{t}\mathrm{d}\lambda = \log\frac{1}{t}.$$

$\square$

**Interpolated $\chi^2$ divergence.** Let $\lambda \in (0,1)$ be a parameter and denote $\bar{\lambda} = 1 - \lambda$. We have,

$$f_{\chi^2,\lambda}(t) = \frac{(t-1)^2}{\lambda t + 1 - \lambda} = f_{\chi^2, 1-\lambda}^*(t).$$

Its derivatives are

$$f_{\chi^2,\lambda}'(t) = \frac{(t-1)(\lambda t + \bar{\lambda} + 1)}{(\lambda t + \bar{\lambda})^2} \quad \text{and} \quad f_{\chi^2,\lambda}''(t) = \frac{2}{(\lambda t + \bar{\lambda})^2}.$$

**Proposition 13.** *For $\lambda \in (0,1)$, the interpolated $\chi^2$-divergence satisfies Assumption 8 with*

$$C_0 = \frac{1}{1-\lambda}, \quad C_0^* = \frac{1}{\lambda}, \quad C_1 = \frac{2}{(1-\lambda)^2}, \quad C_1^* = \frac{2}{\lambda^2}$$

$$C_2 = \frac{4}{27\lambda(1-\lambda)^2}, \quad C_2^* = \frac{4}{27\lambda^2(1-\lambda)}.$$

*Proof.* Note that $0 \ge f_{\chi^2,\lambda}'(0) = -(1+\bar{\lambda})/\bar{\lambda}^2 \ge -2/\bar{\lambda}^2$ is bounded. Since $f_{\chi^2,\lambda}'$ is monotonic increasing with $f_{\chi^2,\lambda}'(1) = 0$, this gives the bound on $C_1$. Next, we bound

$$C_2 = \sup_{t>0} \frac{t}{(\lambda t + \bar{\lambda})^3} = \frac{4}{27\lambda\bar{\lambda}^2},$$

since the supremeum is attained at $t = \bar{\lambda}/(2\lambda)$.

$\square$

**Squared Hellinger distance.** We have,

$$f_H(t) = (1-\sqrt{t})^2 = f_H^*(t), \quad f_H'(t) = 1 - \frac{1}{\sqrt{t}}, \quad f_H''(t) = \frac{1}{2}t^{-3/2}.$$

The squared Hellinger divergence does not satisfy our assumptions since for $t < 1$, $|f_H'(x)| \approx 1/\sqrt{t}$ diverges faster than the $\log 1/t$ rate required by Assumption (A2).

## C.3 Properties and useful lemmas

We state here some useful properties and lemmas that we use throughout the paper.

First, we express the derivatives of $\psi(p,q) = qf(p/q)$ in terms of the derivatives of $f$:

$$\frac{\partial \psi}{\partial p}(p,q) = f'\left(\frac{p}{q}\right) = f^*{}'\left(\frac{q}{p}\right) - \frac{q}{p}(f^*)'\left(\frac{q}{p}\right) \tag{12a}$$

$$\frac{\partial \psi}{\partial q}(p,q) = f\left(\frac{p}{q}\right) - \frac{p}{q}f'\left(\frac{p}{q}\right) = (f^*)'\left(\frac{q}{p}\right) \tag{12b}$$

$$\frac{\partial^2 \psi}{\partial p^2}(p,q) = \frac{1}{q}f''\left(\frac{p}{q}\right) = \frac{q^2}{p^3}(f^*)''\left(\frac{q}{p}\right) \geq 0 \tag{12c}$$

$$\frac{\partial^2 \psi}{\partial q^2}(p,q) = \frac{p^2}{q^3}f''\left(\frac{p}{q}\right) = \frac{1}{p}(f^*)''\left(\frac{q}{p}\right) \geq 0 \tag{12d}$$

$$\frac{\partial^2 \psi}{\partial p \partial q}(p,q) = -\frac{p}{q^2}f''\left(\frac{p}{q}\right) = -\frac{q}{p^2}(f^*)''\left(\frac{q}{p}\right) \leq 0\,, \tag{12e}$$

where the inequalities $f''$, $(f^*)'' \geq 0$ followed from convexity of $f$ and $f^*$ respectively.

The next lemma shows that the function $\psi$ is nearly Lipschitz, up to a log factor. This lemma can be leveraged to directly obtain a bound on statistical error of the $f$-divergence in terms of the expected total variation distance, provided the probabilities are not too small.

**Lemma 14.** *Suppose that $f$ satisfies Assumption 8. Consider $\psi : [0,1] \times [0,1] \to [0,\infty)$ given by $\psi(p,q) = qf(p/q)$. We have, for all $p, p', q, q' \in [0,1]$ with $p \vee p' > 0$, $q \vee q' > 0$, that*

$$|\psi(p',q) - \psi(p,q)| \leq \left(C_1 \max\left\{1, \log \frac{1}{p \vee p'}\right\} + C_0^* \vee C_2\right)|p - p'|$$

$$|\psi(p,q') - \psi(p,q)| \leq \left(C_1^* \max\left\{1, \log \frac{1}{q \vee q'}\right\} + C_0 \vee C_2^*\right)|q - q'|\,.$$

*Proof.* We only prove the first inequality. The second one is identical with the use of $f^*$ rather than $f$. Suppose $p' \geq p$. From the fact that $\psi$ is convex in $p$ together with a Taylor expansion of $\psi(\cdot, q)$ around $p'$, we get,

$$0 \leq \psi(p,q) - \psi(p',q) - (p - p')\frac{\partial \psi}{\partial p}(p',q) = \frac{1}{2}\int_{p'}^{p}\frac{\partial^2 \psi}{\partial p^2}(s,q)(p - s)\mathrm{d}s$$

$$= -\frac{p}{2}\int_{p}^{p'}\frac{\partial^2 \psi}{\partial p^2}(s,q)\mathrm{d}s + \frac{1}{2}\int_{p}^{p'}s\frac{\partial^2 \psi}{\partial p^2}(s,q)\mathrm{d}s$$

$$\leq 0 + C_2(p' - p)\,,$$

where we used $\partial^2 \psi / \partial p^2$ is non-negative due to convexity and, by (12c) and Assumption **(A3)**,

$$s\frac{\partial^2 \psi}{\partial p^2}(s,q) = \frac{s}{q}f''(s/q) \leq 2C_2\,.$$

This yields

$$-(p'-p)\frac{\partial \psi}{\partial p}(p',q) \leq \psi(p,q) - \psi(p',q) \leq -(p'-p)\frac{\partial \psi}{\partial p}(p',q) + C_2(p'-p)\,.$$

We consider two cases based on the sign of $\frac{\partial \psi}{\partial p}(p',q) = f'(p/q)$ (cf. Eq. (12e)).

**Case 1.** $\frac{\partial \psi}{\partial p}(p',q) \geq 0$. Since $q \mapsto f'(p/q)$ is decreasing in $q$, we have

$$0 \leq (p'-p)\frac{\partial \psi}{\partial p}(p',q) = (p'-p)f'(p/q) \leq \lim_{q \to 0}(p'-p)f'(p/q) = (p'-p)f^*(0)\,,$$

where we used $f'(\infty) = f^*(0)$ from Lem. 15. From Assumption **(A1)**, we get the bound

$$|\psi(p,q) - \psi(p',q)| \leq (C_0^* \vee C_2)(p'-p)\,.$$

**Case 2.** $\frac{\partial \psi}{\partial p}(p', q) < 0$. By Assumption **(A2)**, it holds that

$$\left|\frac{\partial \psi}{\partial p}(p', q)\right| \leq C_1 \max\{1, \log(q/p')\} \leq C_1 \max\{1, \log(1/p')\},$$

and thus

$$|\psi(p, q) - \psi(p', q)| \leq \left(C_1 \max\left\{1, \log\frac{1}{p'}\right\} + C_2\right)(p' - p).$$

$\square$

With the above lemma, the estimation error of the empirical $f$-divergence can be upper bounded by the total variation distance between the empirical measure and its population counterpart up to a logarithmic factor, where:

$$\|\hat{P}_n - P\|_{\mathrm{TV}} = \sum_{a \in \mathcal{X}} |\hat{P}_n(a) - P(a)|. \tag{13}$$

Next, we state and prove a technical lemma.

**Lemma 15.** *Suppose the generator $f$ satisfies Assumptions (A1) and (A2). Then,*

$$\lim_{t \to \infty} f'(t) = f^*(0), \quad \text{and} \quad \lim_{t \to \infty} (f^*)'(t) = f(0).$$

*Proof.* We start by observing that

$$\lim_{t \to 0} t|f'(t)| \leq C_1 \lim_{t \to 0} t \vee t \log\frac{1}{t} = 0.$$

Next, a direct calculation gives

$$(f^*)'(1/t) = f(t) - tf'(t),$$

so that taking the limit $t \to 0$ gives

$$\lim_{t \to \infty} (f^*)'(t) = f(0) - \lim_{t \to 0} tf'(t) = f(0).$$

The proof of the other part is identical. $\square$

# D  Plug-in estimator: statistical error

In this section, we prove the high probability concentration bound for the plug-in estimator. There are two keys steps: bounding the statistical error and giving a deviation bound.

Throughout this section, we assume that $P$ and $Q$ are discrete. Let $\{X_i\}_{i=1}^n$ and $\{Y_j\}_{j=1}^m$ be two independent i.i.d. samples from $P$ and $Q$, respectively. We consider the plug-in estimator of the $f$-divergences, i.e., $D_f(\hat{P}_n\|\hat{Q}_m)$. The main results are (a) an upper bound for its statistical error, and (b) a high probability concentration bound. They all hold for the linearized cost $\mathcal{L}_\lambda(\hat{P}_n, \hat{Q}_n)$ and the frontier integral $\mathrm{FI}(\hat{P}_n, \hat{Q}_n)$ due to Prop. 11 and Prop. 12.

## D.1  Statistical error

**Proposition 16.** *Suppose that $f$ satisfies Assumption 8 and $k := |\mathrm{Supp}(P)| \vee |\mathrm{Supp}(Q)| \in \mathbb{N} \cup \{\infty\}$. Let $n, m \geq 3$. Let $c_1 = C_1 + C_1^*$ and $c_2 = C_2 \vee C_0^* + C_2^* \vee C_0$. We have,*

$$\mathbb{E}|D_f(P\|Q) - D_f(\hat{P}_n\|\hat{Q}_m)| \leq (C_1 \log n + C_0^* \vee C_2)\alpha_n(P) + (C_1^* \log m + C_0 \vee C_2^*)\alpha_m(Q) \tag{14}$$

$$+ (C_1 + C_0^* \vee C_2)\beta_n(P) + (C_1^* + C_0 \vee C_2^*)\beta_m(Q),$$

*where $\alpha_n(P) = \sum_{a \in \mathcal{X}} \sqrt{n^{-1}P(a)}$ and $\beta_n(P) = \mathbb{E}\big[\sum_{a:\hat{P}_n(a)=0} P(a) \max\{1, \log(1/P(a))\}\big]$. Furthermore, if $k < \infty$, then*

$$\mathbb{E}|D_f(P\|Q) - D_f(\hat{P}_n\|\hat{Q}_m)| \leq (c_1 \log(n \wedge m) + c_2)\left(\sqrt{\frac{k}{n \wedge m}} + \frac{k}{n \wedge m}\right). \tag{15}$$

The proof relies on two key lemmas—the approximate Lipschitz lemma (Lem. 14) and the missing mass lemma (Lem. 18). The argument breaks into two cases in $P$ (and analogously for $Q$) for each atom $a \in \mathcal{X}$:

(a) $\hat{P}_n(a) > 0$: Since $\hat{P}_n$ is an empirical measure, we have that $\hat{P}_n(a) \geq 1/n$. In this case the approximate Lipschitz lemma gives us the Lipschitzness in $\|P - \hat{P}_n\|_{\text{TV}}$ up to a factor of $\log n$.

(b) $\hat{P}_n(a) = 0$: In this case, the mass corresponding to $P(a)$ is missing in the empirical measure and we directly bound its expectation following similar arguments as in the missing mass literature; see, e.g., [3, 40].

For the first part, we further upper bound the expected total variation distance of the plug-in estimator, which is
$$\|\hat{P}_n - P\|_{\text{TV}} = \sum_{a \in \mathcal{X}} |\hat{P}_n(a) - P(a)| \,.$$

**Lemma 17.** *Assume that $P$ is discrete. For any $n \geq 1$, it holds that*
$$\mathbb{E}\|\hat{P}_n - P\|_{\text{TV}} \leq \alpha_n(P).$$

*Furthermore, if $k = |\text{Supp}(P)| < \infty$, then*
$$\mathbb{E}\|\hat{P}_n - P\|_{\text{TV}} \leq \alpha_n(P) \leq \sqrt{\frac{k}{n}} \,.$$

*Proof.* Using Jensen's inequality, we have,
$$\mathbb{E} \sum_{a \in \text{Supp}(P)} |\hat{P}_n(a) - P(a)| \leq \sum_{a \in \text{Supp}(P)} \sqrt{\mathbb{E}(\hat{P}_n(a) - P(a))^2}$$
$$= \sum_{a \in \text{Supp}(P)} \sqrt{\frac{P(a)(1 - P(a))}{n}} \leq \alpha_n(P) \,,$$

If $k < \infty$, then it follows from Jensen's inequality applied to the concave function $t \mapsto \sqrt{t}$ that
$$\frac{1}{k} \sum_{i=1}^k \sqrt{a_k} \leq \sqrt{\frac{1}{k} \sum_{i=1}^k a_k} \,.$$

Hence, $\alpha_n(P) \leq k/n$ and it completes the proof. $\qquad\square$

For the second part, we treat the missing mass directly.

**Lemma 18** (Missing Mass). *Assume that $k = |\text{Supp}(P)| < \infty$. Then, for any $n \geq 3$,*
$$\mathbb{E}\left[\sum_{a \in \mathcal{X}} \mathbb{1}\{\hat{P}_n(a) = 0\} P(a)\right] \leq \frac{k}{n} \tag{16}$$
$$\beta_n(P) := \mathbb{E}\left[\sum_{a \in \mathcal{X}} \mathbb{1}\{\hat{P}_n(a) = 0\} P(a) \left(1 \vee \log \frac{1}{P(a)}\right)\right] \leq \frac{k \log n}{n} \,, \tag{17}$$

*where $a \vee b := \max\{a, b\}$.*

*Proof.* We prove the second inequality. The first one is identical. Note that $\mathbb{E}[\mathbb{1}\{\hat{P}_n(a) = 0\}] = \mathbb{P}(\hat{P}_n(a) = 0) = (1 - P(a))^n$. Therefore, the left hand side (LHS) of the second inequality is
$$\text{LHS} = \sum_{a \in \mathcal{X}} (1 - P(a))^n P(a) \max\{1, -\log P(a)\}$$
$$\leq \sum_{a \in \mathcal{X}} \frac{1}{n} \vee \frac{\log n}{n} = \frac{k \log n}{n} \,,$$

where we used Lem. 31 and Lem. 32. $\qquad\square$

**Remark 19.** *According to [3, Prop. 3], the bound $k/n$ in (16) is tight up to a constant factor.*

Now, we are ready to prove Prop. 16.

*Proof of Prop. 16.* Define $\Delta_{n,m}(a) := \left|\psi\big(P(a), Q(a)\big) - \psi\big(\hat{P}_n(a), \hat{Q}_m(a)\big)\right|$. We have from the triangle inequality that

$$\Delta_{n,m}(a) \le \underbrace{\left|\psi\big(P(a), Q(a)\big) - \psi\big(\hat{P}_n(a), Q(a)\big)\right|}_{=:\mathcal{T}_1(a)} + \underbrace{\left|\psi\big(\hat{P}_n(a), Q(a)\big) - \psi\big(\hat{P}_n(a), \hat{Q}_m(a)\big)\right|}_{=:\mathcal{T}_2(a)}.$$

Since $\hat{P}_n(a) = 0$ or $\hat{P}_n(a) \ge 1/n$, the approximate Lipschitz lemma (Lem. 14) gives

$$\mathcal{T}_1(a) \le \begin{cases} P(a)\left(C_1 \max\{1, \log(1/P(a))\} + C_0^* \vee C_2\right), & \text{if } \hat{P}_n(a) = 0, \\ |P(a) - \hat{P}_n(a)|\left(C_1 \log n + C_0^* \vee C_2\right), & \text{else.} \end{cases}$$

Consequently, Lem. 17 yields

$$\sum_{a \in \mathcal{X}} \mathbb{E}[\mathcal{T}_1] \le \sum_{a \in \mathcal{X}} \mathbb{E}\left[\mathbb{1}\{\hat{P}_n(a) = 0\}P(a)\left(C_1 \max\{1, \log(1/P(a))\} + C_0^* \vee C_2\right)\right]$$

$$+ \sum_{a \in \mathcal{X}} \mathbb{E}\left[\left|\hat{P}_n(a) - P(a)\right|\right]\left(C_1 \log n + C_0^* \vee C_2\right)$$

$$\le \left(C_1 + C_0^* \vee C_2\right)\beta_n(P) + \left(C_1 \log n + C_0^* \vee C_2\right)\alpha_n(P).$$

Since $\psi(p, q) = qf(p/q) = pf^*(q/p)$, an analogous bound holds for $\mathcal{T}_2$ with the appropriate adjustment of constants. Hence, the inequality (14) holds. Moreover, when $k < \infty$, the inequality (15) follows by invoking again Lem. 18 and Lem. 17. □

Invoking Prop. 10 and Prop. 16 for the interpolated KL divergence leads to the following result.

**Proposition 20.** *Assume that $k = |\mathrm{Supp}(P)| \vee |\mathrm{Supp}(Q)| < \infty$. For any $\lambda \in (0, 1)$, it holds that*

$$\mathbb{E}\left|\mathrm{KL}_\lambda(\hat{P}_n\|\hat{Q}_m) - \mathrm{KL}_\lambda(P\|Q)\right|$$

$$\le \left[\left(1 + \frac{(1-\lambda)^2}{\lambda}\right)\log(n \wedge m) + \left(\log\frac{1}{\lambda} - 1 + \lambda\right) \vee \frac{1}{2} + (1-\lambda) \vee \frac{1-\lambda}{8\lambda}\right]$$

$$\times \left(\sqrt{\frac{k}{n \wedge m}} + \frac{k}{n \wedge m}\right).$$

*Moreover, for any $\lambda_{n,m} \in (0, 1/2)$,*

$$\mathbb{E}\left[\sup_{\lambda \in [\lambda_{n,m}, 1-\lambda_{n,m}]}\left\{\left|\mathrm{KL}_\lambda(\hat{P}_n\|\hat{Q}_m) - \mathrm{KL}_\lambda(P\|Q)\right| + \left|\mathrm{KL}_{1-\lambda}(\hat{Q}_m\|\hat{P}_n) - \mathrm{KL}_{1-\lambda}(Q\|P)\right|\right\}\right]$$

$$\le 2\left((1 + 1/\lambda_{n,m})\log n + \log\frac{1}{\lambda_{n,m}} \vee \frac{1}{2} + 1 \vee \frac{1}{8\lambda_{n,m}}\right)\left(\sqrt{\frac{k}{n \wedge m}} + \frac{k}{n \wedge m}\right).$$

*Proof.* We only prove the second inequality. The first one is a direct consequence of Prop. 10 and Prop. 16. From the proof of Prop. 16 we have

$$\left|\mathrm{KL}_\lambda(\hat{P}_n\|\hat{Q}_m) - \mathrm{KL}_\lambda(P\|Q)\right|$$

$$\le \sum_{a \in \mathcal{X}} \mathbb{1}\{\hat{P}_n(a) = 0\}P(a)\left(C_1 \max\{1, \log(1/P(a))\} + C_0^* \vee C_2\right)$$

$$+ \sum_{a \in \mathcal{X}} \mathbb{1}\{\hat{Q}_m(a) = 0\}Q(a)\left(C_1^* \max\{1, \log(1/Q(a))\} + C_0 \vee C_2^*\right)$$

$$+ \sum_{a \in \mathcal{X}} \left|P(a) - \hat{P}_n(a)\right|\left(C_1 \log n + C_0^* \vee C_2\right) + \sum_{a \in \mathcal{X}} \left|Q(a) - \hat{Q}_m(a)\right|\left(C_1^* \log m + C_0 \vee C_2^*\right).$$

Note that, for the intepolated KL divergence, we have

$$C_0 = 1 - \lambda \le 1, \quad C_0^* = \log \frac{1}{\lambda} - 1 + \lambda \le \log \frac{1}{\lambda_{n,m}}$$

$$C_1 = 1, \quad C_1^* = \frac{(1-\lambda)^2}{\lambda} \le \frac{1}{\lambda_{n,m}}$$

$$C_2 = 1/2, \quad C_2^* = \frac{1-\lambda}{8\lambda} \le \frac{1}{8\lambda_{n,m}}$$

for all $\lambda \in [\lambda_{n,m}, 1 - \lambda_{n,m}]$. The claim then follows from the same steps of Prop. 16. $\quad\square$

## D.2 Concentration bound

We now state and prove the concentration bound for general $f$-divergences which satisfy our regularity assumptions. We start by considering concentration around the expectation.

**Proposition 21.** *Consider the $f$-divergence $D_f$ where $f$ satisfies Assumptions (A1)-(A3). For any $t > 0$ and any dicrete distributions $P, Q$, we have,*

$$\mathbb{P}\left(|D_f(\hat{P}_n\|\hat{Q}_m) - \mathbb{E}[D_f(\hat{P}_n\|\hat{Q}_m)]| > \varepsilon\right) \le 2\exp\left(-\frac{(n \wedge m)\varepsilon^2}{2(c_1 \log(n \wedge m) + c_2)^2}\right),$$

*where $c_1 = C_1 + C_1^*$ and $c_2 = C_2 \vee C_0^* + C_2^* \vee C_0$.*

*Proof.* We first establish that $D_f$ satisfies the bounded deviation property and then invoke McDiarmid's inequality.

We start with some notation. As before, define $\psi(p,q) = qf(p/q)$. Without loss of generality, let $\mathcal{X} = \text{Supp}(P) \cup \text{Supp}(Q)$. Define the function $\Phi : \mathcal{X}^{n+m} \to \mathbb{R}$ so that

$$\Phi(X_1, \cdots, X_n, Y_1, \cdots, Y_m) = D_f(\hat{P}_n\|\hat{Q}_m).$$

We now show the bounded deviation property of $\Phi$. Fix some $T = (x_1, \cdots, x_n, y_1, \cdots, y_m) \in \mathcal{X}^{n+m}$ and let $T' = (x_1', \cdots, x_n', y_1', \cdots, y_m') \in \mathcal{X}^{n+m}$ be such that $T$ and $T'$ differ only on $x_i = a \ne a' = x_i'$. Suppose the number of occurrences of $a$ in the $x$-component of $T$ is $l$ and of $a'$ is $l'$, while their corresponding $y$-components are $mq$ and $mq'$ respectvely. We now have

$$|\Phi(T') - \Phi(T)| = \left|\psi\left(\frac{s-1}{n}, q\right) - \psi\left(\frac{s}{n}, q\right) + \psi\left(\frac{s'+1}{n}, q'\right) - \psi\left(\frac{s'}{n}, q'\right)\right|$$

$$\le \left|\psi\left(\frac{s-1}{n}, q\right) - \psi\left(\frac{s}{n}, q\right)\right| + \left|\psi\left(\frac{s'+1}{n}, q'\right) - \psi\left(\frac{s'}{n}, q'\right)\right|$$

$$\le \frac{2}{n}(C_1 \log n + C_0^* \vee C_2) =: B_i,$$

where we used the triangle inequality first and then invoked Lem. 14. Likewise, if $A$ and $A'$ differ only in $y_i$ and $y_i'$, an analogous argument gives

$$|\Phi(T') - \Phi(T)| \le \frac{2}{m}(C_1^* \log m + C_0 \vee C_2^*) =: B_i^*.$$

With this we can use McDiarmid's inequality (cf. Thm. 29) to bound

$$\mathbb{P}\left(|D_f(\hat{P}_n\|\hat{Q}_m) - \mathbb{E}[D_f(\hat{P}_n\|\hat{Q}_m)]| > \varepsilon\right) \le h(\varepsilon),$$

where

$$h(\varepsilon) = 2\exp\left(-\frac{2\varepsilon^2}{\sum_{i=1}^n B_i^2 + \sum_{i=n+1}^{n+m}(B_i^*)^2}\right) \le 2\exp\left(-\frac{(n \wedge m)\varepsilon^2}{2(c_1 \log(n \wedge m) + c_2)^2}\right).$$

$\square$

Hence, the concentration bound around the population $f$-divergence follows directly from Prop. 16 and Prop. 21.

**Theorem 22.** *Assume that $P$ and $Q$ are discrete and let $k = |\mathrm{Supp}(P)| \vee |\mathrm{Supp}(Q)| \in \mathbb{N} \cup \{\infty\}$. For any $\delta \in (0,1)$, it holds that, with probability at least $1 - \delta$,*

$$\left| D_f(\hat{P}_n \| \hat{Q}_m) - D_f(P \| Q) \right| \leq \left( c_1 \log(n \wedge m) + c_2 \right) \sqrt{\frac{2}{n \wedge m} \log \frac{2}{\delta}}$$
$$+ \left( C_1 \log n + C_0^* \vee C_2 \right) \alpha_n(P) + \left( C_1^* \log m + C_0 \vee C_2^* \right) \alpha_m(Q)$$
$$+ \left( C_1 + C_0^* \vee C_2 \right) \beta_n(P) + \left( C_1^* + C_0 \vee C_2^* \right) \beta_m(Q) \,.$$

*Furthermore, if $k < \infty$, then, with probability at least $1 - \delta$,*

$$\left| D_f(\hat{P}_n \| \hat{Q}_m) - D_f(P \| Q) \right| \leq \left( c_1 \log(n \wedge m) + c_2 \right) \left( \sqrt{\frac{2}{n \wedge m} \log \frac{2}{\delta}} + \sqrt{\frac{k}{n \wedge m}} + \frac{k}{n \wedge m} \right) \,.$$

*Proof of Thm. 22.* We only prove the second inequality. The first one follows from a similar argument. According to Prop. 16, we have

$$\left| D_f(\hat{P}_n \| \hat{Q}_m) - \mathbb{E}[D_f(\hat{P}_n \| \hat{Q}_m)] \right|$$
$$\geq \left| D_f(\hat{P}_n \| \hat{Q}_m) - D_f(P \| Q) \right| - \left| \mathbb{E}[D_f(\hat{P}_n \| \hat{Q}_m)] - D_f(P \| Q) \right|$$
$$\geq \left| D_f(\hat{P}_n \| \hat{Q}_m) - D_f(P \| Q) \right| - \left( c_1 \log(n \wedge m) + c_2 \right) \left( \sqrt{\frac{k}{n \wedge m}} + \frac{k}{n \wedge m} \right) \,.$$

By Prop. 21, it holds that

$$\mathbb{P}\left( \left| D_f(\hat{P}_n \| \hat{Q}_m) - D_f(P \| Q) \right| > \varepsilon + \left( c_1 \log(n \wedge m) + c_2 \right) \left( \sqrt{\frac{k}{n \wedge m}} + \frac{k}{n \wedge m} \right) \right) \leq h(\epsilon) \,,$$

where

$$h(\epsilon) = 2 \exp\left( -\frac{(n \wedge m)\varepsilon^2}{2(c_1 \log(n \wedge m) + c_2)^2} \right) \,.$$

The claim then follows from setting $h(\epsilon) = \delta$ and solving for $\epsilon$. $\qquad\square$

## E   Add-constant smoothing: statistical error

In this section, we apply add-constant smoothing to estimate the $f$-divergences and study its statistical error. All the results hold for the linearized cost $\mathcal{L}_\lambda(\hat{P}_n, \hat{Q}_n)$ and the frontier integral $\mathrm{FI}(\hat{P}_n, \hat{Q}_n)$ due to Prop. 11 and Prop. 12.

For notational simplicity, we assume that $P$ and $Q$ are supported on a common finite alphabet with size $k < \infty$. Without loss of generality, let $\mathcal{X}$ be the support. Consider $P \in \mathcal{P}(\mathcal{X})$ and an i.i.d. sample $\{X_i\}_{i=1}^n \sim P$. The add-constant estimator of $P$ is defined by

$$\hat{P}_{n,b}(a) = \frac{N_a + b}{n + kb}, \quad \text{for all } a \in \mathcal{X} \,,$$

where $b > 0$ is a constant and $N_a = |\{i \in [n] : X_i = a\}|$ is the number of times the symbol $a$ appears in the sample. In practice, $b = b_a$ could be different depending on the value of $N_a$, but we use the same constant $b$ for simplicity. Similarly, We define $\hat{Q}_{m,b}$ with $M_a = |\{i \in [m] : Y_i = a\}|$. The goal is to upper bound the statistical error

$$\mathbb{E}\left| D_f(P \| Q) - D_f(\hat{P}_{n,b} \| \hat{Q}_{m,b}) \right| \tag{18}$$

under Assumption 8.

Compared to the statistical error of the plug-in estimator, a key difference is that each entry in the add-constant estimator is at least $(n + kb)^{-1} \wedge (m + kb)^{-1}$. Hence, we can directly apply the approximate Lipschitz lemma without the need to control the missing mass part. Another difference is that the total variation distance is now between the add-constant estimator and its population counterpart, which can be bounded as follows.

**Lemma 23.** *Assume that $k = \mathrm{Supp}(P) < \infty$. Then, for any $b > 0$,*

$$\sum_{a \in \mathcal{X}} \mathbb{E} \left| \hat{P}_{n,b}(a) - P(a) \right| \leq \sum_{a \in \mathcal{X}} \frac{\sqrt{nP(a)(1 - P(a))} + bk \left| P(a) - 1/k \right|}{n + kb} \leq \frac{\sqrt{kn} + 2b(k-1)}{n + kb}.$$

*Proof.* Note that

$$\left| \hat{P}_{n,b}(a) - P(a) \right| = \left| \frac{N_a - nP(a)}{n + kb} + \frac{b(1 - kP(a))}{n + kb} \right| \leq \left| \frac{N_a - nP(a)}{n + kb} \right| + \left| \frac{b(1 - kP(a))}{n + kb} \right|.$$

Using Jensen's inequality, we have

$$\sum_{a \in \mathcal{X}} \mathbb{E} \left| \hat{P}_{n,b}(a) - P(a) \right| \leq \sum_{a \in \mathcal{X}} \left[ \sqrt{ \mathbb{E} \left| \frac{N_a - nP(a)}{n + kb} \right|^2 } + \frac{c \left| 1 - kP(a) \right|}{n + kb} \right]$$

$$= \sum_{a \in \mathcal{X}} \left[ \frac{\sqrt{nP(a)(1 - P(a))}}{n + kb} + \frac{bk \left| 1/k - P(a) \right|}{n + kb} \right].$$

We claim that

$$\sum_{a \in \mathcal{X}} \left| P(a) - \frac{1}{k} \right| \leq \frac{2(k-1)}{k}.$$

If this is true, we have

$$\sum_{a \in \mathcal{X}} \mathbb{E} \left| \hat{P}_{n,b}(a) - P(a) \right| \leq \frac{\sqrt{kn} + 2b(k-1)}{n + kb},$$

since $\sum_{a \in \mathcal{X}} \sqrt{P(a)(1 - P(a))} \leq \sqrt{k}$ It then remains to prove the claim. Take $a_1, a_2 \in \mathcal{X}$ such that $P(a_1) \geq k^{-1} \geq P(a_2)$. It is clear that

$$\left| P(a_1) - \frac{1}{k} \right| + \left| P(a_2) - \frac{1}{k} \right| \leq \left| P(a_1) + P(a_2) - \frac{1}{k} \right| + \left| P(a_2) - P(a_2) - \frac{1}{k} \right|$$

$$= P(a_1) + P(a_2).$$

Repeating this argument gives

$$\sum_{a \in \mathcal{X}} \left| P(a) - \frac{1}{k} \right| \leq 1 - \frac{1}{k} + \frac{k-1}{k} = \frac{2(k-1)}{k}.$$

$\square$

The next proposition gives the upper bound for the statistical error of the add-constant estimator.

**Proposition 24.** *Suppose that $f$ satisfies Assumption 8 and $k = |\mathcal{X}| < \infty$. We have, for any $n, m \geq 3$,*

$$\mathbb{E} \left| D_f(P\|Q) - D_f(\hat{P}_{n,b} \| \hat{Q}_{m,b}) \right| \leq \left[ \frac{n\alpha_n(P)}{n + kb} + \gamma_{n,k}(P) \right] \left( C_1 \log(n/b + k) + C_0^* \vee C_2 \right)$$

$$+ \left[ \frac{m\alpha_m(Q)}{m + kb} + \gamma_{m,k}(Q) \right] \left( C_1^* \log(m/b + k) + C_0 \vee C_2^* \right)$$

$$\leq \left( C_1 \log(n/b + k) + C_0^* \vee C_2 \right) \frac{\sqrt{kn} + 2b(k-1)}{n + kb}$$

$$+ \left( C_1^* \log(m/b + k) + C_0 \vee C_2^* \right) \frac{\sqrt{km} + 2b(k-1)}{m + kb},$$

*where $\gamma_{n,k}(P) = (n + bk)^{-1} bk \sum_{a \in \mathcal{X}} |P(a) - 1/k|$.*

*Proof.* Following the proof of Prop. 16, we define

$$\Delta_{n,m}(a) := \left| \psi(P(a), Q(a)) - \psi(\hat{P}_{n,b}(a), \hat{Q}_{m,b}(a)) \right| .$$

We have from the triangle inequality that

$$\Delta_{n,m}(a) \leq \underbrace{\left| \psi(P(a), Q(a)) - \psi(\hat{P}_{n,b}(a), Q(a)) \right|}_{=:\mathcal{T}_1(a)} + \underbrace{\left| \psi(\hat{P}_{n,b}(a), Q(a)) - \psi(\hat{P}_{n,b}(a), \hat{Q}_{m,b}(a)) \right|}_{=:\mathcal{T}_2(a)} .$$

Since $\hat{P}_{n,b}(a) \geq b/(n + kb)$, the approximate Lipschitz lemma (Lem. 14) gives

$$\mathcal{T}_1(a) \leq |P(a) - \hat{P}_{n,b}(a)| \left( C_1 \log(n/b + k) + C_0^* \vee C_2 \right),$$

By Lem. 23, it holds that

$$\frac{\sum_{a \in \mathcal{X}} \mathbb{E}[\mathcal{T}_1(a)]}{C_1 \log(n/b + k) + C_0^* \vee C_2} \leq \sum_{a \in \mathcal{X}} \left[ \frac{\sqrt{nP(a)}}{n + kb} + \frac{bk\,|1/k - P(a)|}{n + kb} \right] = \frac{n\alpha_n(P)}{n + kb} + \gamma_{n,k}(P)$$

$$\leq \frac{\sqrt{kn} + 2b(k - 1)}{n + kb} .$$

Since $\psi(p, q) = qf(p/q) = pf^*(q/p)$, an analogous bound holds for $\mathcal{T}_2(a)$ with the appropriate adjustment of constants and the sample size. Putting these together, we get,

$$\mathbb{E}\left| D_f(P\|Q) - D_f(\hat{P}_{n,b}\|\hat{Q}_{m,b}) \right| \leq \mathbb{E}\left[ \sum_{a \in \mathcal{X}} |\Delta_n(a)| \right]$$

$$\leq \left[ \frac{n\alpha_n(P)}{n + kb} + \gamma_{n,k}(P) \right] \left( C_1 \log(n/b + k) + C_0^* \vee C_2 \right)$$

$$+ \left[ \frac{m\alpha_m(Q)}{m + kb} + \gamma_{m,k}(Q) \right] \left( C_1^* \log(m/b + k) + C_0 \vee C_2^* \right)$$

$$\leq \left( C_1 \log(n/b + k) + C_0^* \vee C_2 \right) \frac{\sqrt{kn} + 2b(k - 1)}{n + kb}$$

$$+ \left( C_1^* \log(m/b + k) + C_0 \vee C_2^* \right) \frac{\sqrt{km} + 2b(k - 1)}{m + kb} .$$

$\square$

The concentration bound for the add-constant estimator can be proved similarly.

## F   Quantization error

In this section, we study the quantization error of $f$-divergences, i.e.,

$$\inf_{|\mathcal{S}| \leq k} |D_f(P\|Q) - D_f(P_{\mathcal{S}}\|Q_{\mathcal{S}})|, \tag{19}$$

where the infimum is over all partitions of $\mathcal{X}$ of size no larger than $k$, and $P_{\mathcal{S}}$ and $Q_{\mathcal{S}}$ are the quantized versions of $P$ and $Q$ according to $\mathcal{S}$, respectively. Note that we do not assume $\mathcal{X}$ to be discrete in this section. All the results hold for the linearized cost $\mathcal{L}_\lambda(\hat{P}_n, \hat{Q}_n)$ and the frontier integral $\mathrm{FI}(\hat{P}_n, \hat{Q}_n)$ due to Prop. 11 and Prop. 12.

Our analysis is inspired by the following result, which shows that the $f$-divergence can be approximated by its quantized counterpart; see, e.g., [22, Theorem 6].

**Theorem 25.** *For any $P, Q \in \mathcal{P}(\mathcal{X})$, it holds that*

$$D_f(P\|Q) = \sup_{\mathcal{S}} D_f(P_{\mathcal{S}}\|Q_{\mathcal{S}}), \tag{20}$$

*where the supremum is over all finite partitions of $\mathcal{X}$.*

The next theorem holds for general $f$-divergences without the requirement of Assumption 8.

**Theorem 26.** *For any $k \geq 1$, we have*

$$\sup_{P,Q} \inf_{|\mathcal{S}| \leq 2k} |D_f(P\|Q) - D_f(P_\mathcal{S}\|Q_\mathcal{S})| \leq \frac{f(0) + f^*(0)}{k}.$$

*Proof.* Assume $f(0) + f^*(0) < \infty$. Otherwise, there is nothing to prove. Fix two distributions $P, Q$ over $\mathcal{X}$. Partition the measurable space $\mathcal{X}$ into

$$\mathcal{X}_1 = \left\{ x \in \mathcal{X} : \frac{\mathrm{d}P}{\mathrm{d}Q}(x) \leq 1 \right\}, \quad \text{and,} \quad \mathcal{X}_2 = \left\{ x \in \mathcal{X} : \frac{\mathrm{d}P}{\mathrm{d}Q}(x) > 1 \right\},$$

so that

$$D_f(P\|Q) = \int_{\mathcal{X}_1} f\left(\frac{\mathrm{d}P}{\mathrm{d}Q}(x)\right) \mathrm{d}Q(x) + \int_{\mathcal{X}_2} f^*\left(\frac{\mathrm{d}Q}{\mathrm{d}P}(x)\right) \mathrm{d}P(x) =: D_f^+(P\|Q) + D_{f^*}^+(Q\|P).$$

We quantize $\mathcal{X}_1$ and $\mathcal{X}_2$ separately, starting with $\mathcal{X}_1$. Define sets $S_1, \cdots, S_k$ as

$$S_m = \left\{ x \in \mathcal{X}_1 : \frac{f(0)(m-1)}{k} \leq f\left(\frac{\mathrm{d}P}{\mathrm{d}Q}(x)\right) < \frac{f(0)m}{k} \right\},$$

where the last set $S_k$ is also extended to include $\{x \in \mathcal{X}_1 : f((\mathrm{d}P/\mathrm{d}Q)(x)) = f(0)\}$. Since $f$ is nonincreasing on $(0, 1]$, it follows that $\sup_{x \in \mathcal{X}_1} f((\mathrm{d}P/\mathrm{d}Q)(x)) \leq f(0)$. As a result, the collection $\mathcal{S} = \{S_1, \cdots, S_k\}$ is a partition of $\mathcal{X}_1$. This gives

$$\frac{f(0)}{k} \sum_{m=1}^{k} (m-1) Q[S_m] \leq D_f^+(P\|Q) \leq \frac{f(0)}{k} \sum_{m=1}^{k} m\, Q[S_m]. \tag{21}$$

Further, since $f$ is nonincreasing on $(0, 1]$, we also have

$$\frac{f(0)(m-1)}{k} \leq f\left(\sup_{x \in F_m} \frac{\mathrm{d}P}{\mathrm{d}Q}(x)\right) \leq f\left(\frac{P[F_m]}{Q[F_m]}\right) \leq f\left(\inf_{x \in F_m} \frac{\mathrm{d}P}{\mathrm{d}Q}(x)\right) \leq \frac{f(0)m}{k}.$$

Hence, it follows that

$$\frac{f(0)}{k} \sum_{m=1}^{k} (m-1) Q[S_m] \leq D_f^+(P_{\mathcal{S}_1}\|Q_{\mathcal{S}_1}) \leq \frac{f(0)}{k} \sum_{m=1}^{k} m\, Q[S_m]. \tag{22}$$

Putting (21) and (22) together gives

$$\inf_{|\mathcal{S}_1| \leq k} \left| D_f^+(P\|Q) - D_f^+(P_{\mathcal{S}_1}\|Q_{\mathcal{S}_1}) \right| \leq \frac{f(0)}{k} \sum_{m=1}^{k} Q[S_m] \leq \frac{f(0)}{k}, \tag{23}$$

since $\sum_{m=1}^{k} Q[S_m] = Q[\mathcal{X}_1] \leq 1$. Repeating the same argument with $P$ and $Q$ interchanged and replacing $f$ by $f^*$ gives

$$\inf_{|\mathcal{S}_2| \leq k} \left| D_{f^*}^+(Q\|P) - D_{f^*}^+(Q_{\mathcal{S}_2}\|P_{\mathcal{S}_2}) \right| \leq \frac{f^*(0)}{k}. \tag{24}$$

To complete the proof, we upper bound the inf of $\mathcal{S}$ over all partitions of $\mathcal{X}$ with $|\mathcal{S}| = k$ by the inf over $\mathcal{S} = \mathcal{S}_1 \cup \mathcal{S}_2$ with partitions $\mathcal{S}_1$ of $\mathcal{X}_1$ and $\mathcal{S}_2$ of $\mathcal{X}_2$, and $|\mathcal{S}_1| = |\mathcal{S}_2| = k$. Now, under this partitioning, we have, $D_f^+(P_\mathcal{S}\|Q_\mathcal{S}) = D_f^+(P_{\mathcal{S}_1}\|Q_{\mathcal{S}_1})$ and $D_{f^*}^+(Q_\mathcal{S}\|P_\mathcal{S}) = D_{f^*}^+(Q_{\mathcal{S}_2}\|P_{\mathcal{S}_2})$. Putting this together with the triangle inequality, we get,

$$\inf_{|\mathcal{S}| \leq 2k} \left| D_f(P\|Q) - D_f(P_\mathcal{S}\|Q_\mathcal{S}) \right|$$

$$\leq \inf_{\mathcal{S}=\mathcal{S}_1 \cup \mathcal{S}_2} \left\{ \left| D_f^+(P\|Q) - D_f^+(P_\mathcal{S}\|Q_\mathcal{S}) \right| + \left| D_{f^*}^+(Q\|P) - D_{f^*}^+(Q_\mathcal{S}\|P_\mathcal{S}) \right| \right\}$$

$$= \inf_{|\mathcal{S}_1| \leq k} \left| D_f^+(P\|Q) - D_f^+(P_{\mathcal{S}_1}\|Q_{\mathcal{S}_1}) \right| + \inf_{|\mathcal{S}_2| \leq k} \left| D_{f^*}^+(Q\|P) - D_{f^*}^+(Q_{\mathcal{S}_2}\|P_{\mathcal{S}_2}) \right|$$

$$\leq \frac{f(0) + f^*(0)}{k}.$$

$\square$

Table 2: Add-constant estimators.

| Braess-Sauer | Krichevsky-Trofimov | Laplace |
|---|---|---|
| $b_a = 1/2$ if $a$ does not appear | | |
| $b_a = 1$ if $a$ appears once | $b \equiv 1/2$ | $b \equiv 1$ |
| $b_a = 3/4$ if $a$ appears more than once | | |

Now, combining Prop. 16 and Thm. 26 leads to an upper bound for the overall estimation error.

**Theorem 27.** *Let $\mathcal{S}_k$ be a partition of $\mathcal{X}$ such that $|\mathcal{S}| = k \geq 2$ and its quantization error satisfies the bound in Thm. 26, i.e.,*

$$|D_f(P\|Q) - D_f(P_{\mathcal{S}_k}\|Q_{\mathcal{S}_k})| \leq \frac{f(0) + f^*(0)}{k}.$$

*Then, for any $n, m \geq 3$,*

$$\mathbb{E}\left|D_f(\hat{P}_{\mathcal{S}_k,n}\|\hat{Q}_{\mathcal{S}_k,m}) - D_f(P\|Q)\right|$$
$$\leq (C_1 \log n + C_0^* \vee C_2)\alpha_n(P) + (C_1^* \log m + C_0 \vee C_2^*)\alpha_m(Q)$$
$$+ (C_1 + C_0^* \vee C_2)\beta_n(P) + (C_1^* + C_0 \vee C_2^*)\beta_m(Q) + \frac{f(0) + f^*(0)}{k}$$
$$\leq (c_1 \log(n \wedge m) + c_2)\left(\sqrt{\frac{k}{n \wedge m}} + \frac{k}{n \wedge m}\right) + \frac{f(0) + f^*(0)}{k},$$

*where $c_1 = C_1 + C_1^*$ and $c_2 = C_2 \vee C_0^* + C_2^* \vee C_0$.*

According to Thm. 27, a good choice of quantization level $k$ is of order $\Theta(n^{1/3})$ which balances between the two types of errors.

## G   Experimental details

We investigate the empirical behavior of the divergence frontier and the frontier integral on both synthetic and real data. Our main findings are: 1) the statistical error bound is tight—it approximately reveals the rate of convergence of the plug-in estimator. 2) The smoothed distribution estimators improve the estimation accuracy. For simplicity, we consider $m = n$ throughout this section.

**Performance Metric.** We are interested in the estimation of the divergence frontier $\mathcal{F}(P, Q)$ and the frontier integral $\text{FI}(P, Q)$ using estimators $\mathcal{F}(\hat{P}_n, \hat{Q}_n)$ and $\text{FI}(\hat{P}_n, \hat{Q}_n)$, respectively. We measure the quality of estimation using the absolute error, which is defined as

$$\sup_{\lambda \in [0.01, 0.99]} \left\{\left|\text{KL}(\hat{P}_n\|\hat{R}_\lambda) - \text{KL}(P\|R)\right| + \left|\text{KL}(\hat{Q}_n\|\hat{R}_\lambda) - \text{KL}(Q\|R)\right|\right\}$$

for the divergence frontier (cf. Cor. 2 with $\lambda_0 = 0.01$), and, $|\text{FI}(\hat{P}_n, \hat{Q}_n) - \text{FI}(P, Q)|$ for the frontier integral. Here $\hat{R}_\lambda := \lambda\hat{P}_n + (1 - \lambda)\hat{Q}_n$. For the real data, we measure the error of estimating $\mathcal{F}(P_{\mathcal{S}_k}, Q_{\mathcal{S}_k})$ by $\mathcal{F}(\hat{P}_{\mathcal{S}_k,n}, \hat{Q}_{\mathcal{S}_k,n})$ and similarly for FI. The results for the divergence frontier is almost identical to the result for the frontier integral. We present both of them in the plots but focus on the latter in the text.

### G.1   Synthetic data

We focus on the case when the support is finite and illustrate the statistical behavior of the Frontier Integral on synthetic data.

**Settings.**   Let $k = |\mathcal{X}|$ be the support size. Following the experimental settings in [47], we consider three types of distributions: 1) the Zipf$(r)$ distribution with $r \in \{0, 1, 2\}$ where $P(i) \propto i^{-r}$. Note that Zipf$(r)$ is regularly varying with index $-r$; see, e.g., [56, Appendix B]. 2) the Step distribution where $P(i) = 1/2$ for the first half bins and $P(i) = 3/2$ for the second half bins. 3) the Dirichlet

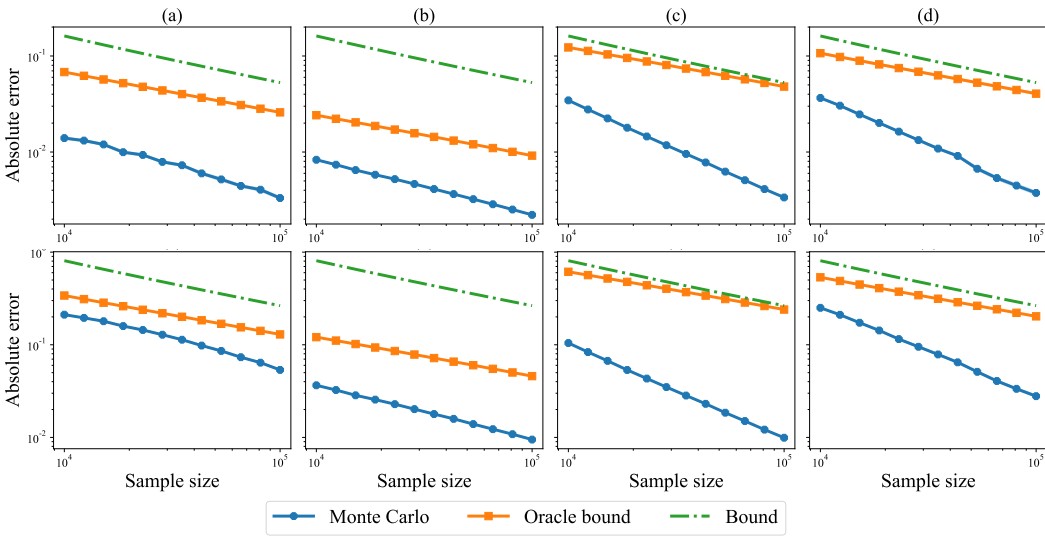

Figure 9: Statistical error versus sample size on synthetic data with $k = 10^3$ (log-log scale) for the frontier integral **(top)** and the divergence frontier **(bottom)**. **(a)**: Zipf(2) and Dir(**1**); **(b)**: Zipf(2) and Zipf(2); **(c)**: Zipf(0) and Zipf(0); **(d)**: Dir(**1**) and Dir(**1/2**).

distribution $\mathrm{Dir}(\alpha)$ with $\alpha \in \{\mathbf{1/2}, \mathbf{1}\}$. In total, there are 6 different distributions. Since the Frontier Integral is symmetric, there are 21 different pairs of $(P, Q)$. For each pair $(P, Q)$, we generate i.i.d. samples of size $n$ from each of them, and then compute the absolute error. We repeat the process 100 times and report its mean and standard error, which is referred to as the Monte Carlo estimate of the expected absolute error.

**Statistical error.** To study the tightness of the statistical error bounds (5), we compare both the distribution-free bound ("Bound") and the distribution-dependent bound ("Oracle bound") with the Monte Carlo estimate ("Monte Carlo"). We call the distribution-free bound the "bound" and the distribution-dependent bound the "oracle bound". We consider three different experiments. First, we fix the support size $k = 10^3$ and increase the sample size $n$ from $10^3$ to $10^4$. Second, we fix $n = 2 \times 10^4$ and increase $k$ from 10 to $10^4$. Third, we fix $k = 10^3$ and $n = 10^4$, and set $Q$ to be the Zipf($r$) with $r$ ranging from 0 to 2. For each of these experiments, we give four typical plots among all pairs of distributions we consider. Note that the two bounds are divided by the same constant for the sake of comparison.

As shown in Fig. 9, the two bounds decreases with $n$ at a similar rate. The oracle bound demonstrates the largest improvement compared to the bound when both $P$ and $Q$ have fast-decaying tails (i.e., with index $-2$). In some cases, the Monte Carlo estimate demonstrates a similar rate of convergence as the bounds; while, in other cases, the Monte Carlo estimate can have a faster rate. This suggests that the bound (5) is at least close to being tight up to a multiplicative constant.

Fig. 10 shows that the oracle bound increases with $k$ at a slower rate than the one of the bound. In fact, it is much slower when both $P$ and $Q$ decay fast. For the Monte Carlo estimate, it can have either a slower or faster rate than the bound depending on the underlying distributions.

The results for the third experiment is in Fig. 11. While the bound remains the same for different tails of $Q$, the oracle bound is adapted to the decaying index of $Q$. The absolute error of the Monte Carlo estimate is usually increasing in the beginning and then decreasing after some threshold.

**Distribution estimators.** We then compare 4 different distribution estimators with the empirical measures ("Empirical") as discussed in [47]. For each $a \in \mathcal{X}$, let $n_a$ be the number of times $a$ appears in the sample $\{X_i\}_{i=1}^n$ and let $\varphi_t$ be the number of symbols appearing $t$ times in the sample. The *(modified) Good-Turing* estimator is defined as $\hat{P}_{\mathrm{GT},n}(a) \propto n_a$ if $n_a > \varphi_{n_a+1}$ and $\hat{P}_{\mathrm{GT},n}(a) \propto [\varphi_{n_a+1} + 1](n_a + 1)/\varphi_{n_a}$ otherwise. The remaining three estimators are all based on the add-$b$ smoothing introduced in Sec. 3. For the *Braess-Sauer* estimator, the parameter $b = b_a$ is

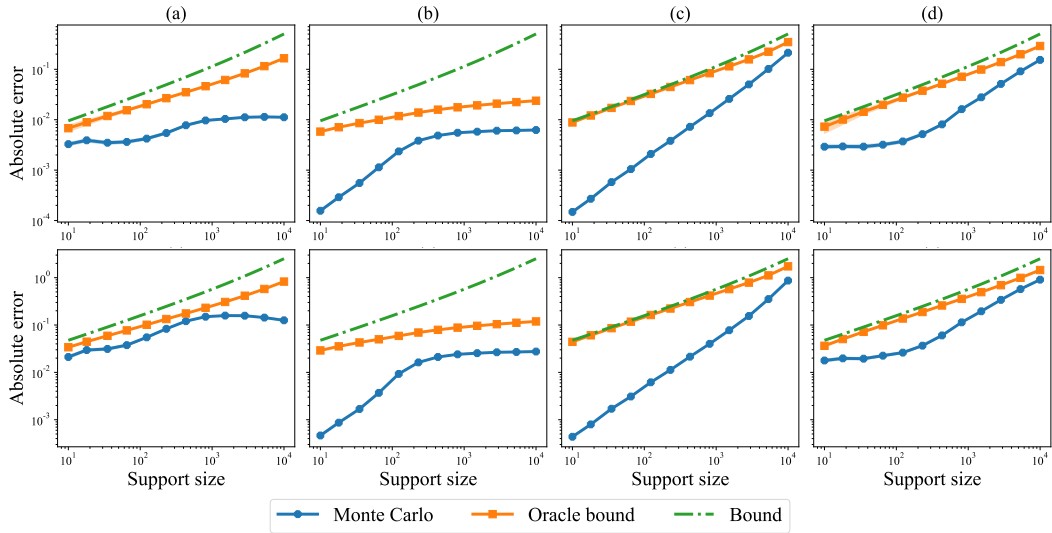

Figure 10: Statistical error versus support size on synthetic data with $n = 2 \times 10^4$ (log-log scale) for the frontier integral **(top)** and the divergence frontier **(bottom)**. **(a)**: $\mathrm{Zipf}(2)$ and $\mathrm{Dir}(\mathbf{1})$; **(b)**: $\mathrm{Zipf}(2)$ and $\mathrm{Zipf}(2)$; **(c)**: $\mathrm{Zipf}(0)$ and $\mathrm{Zipf}(0)$; **(d)**: $\mathrm{Dir}(\mathbf{1})$ and $\mathrm{Dir}(\mathbf{1}/2)$.

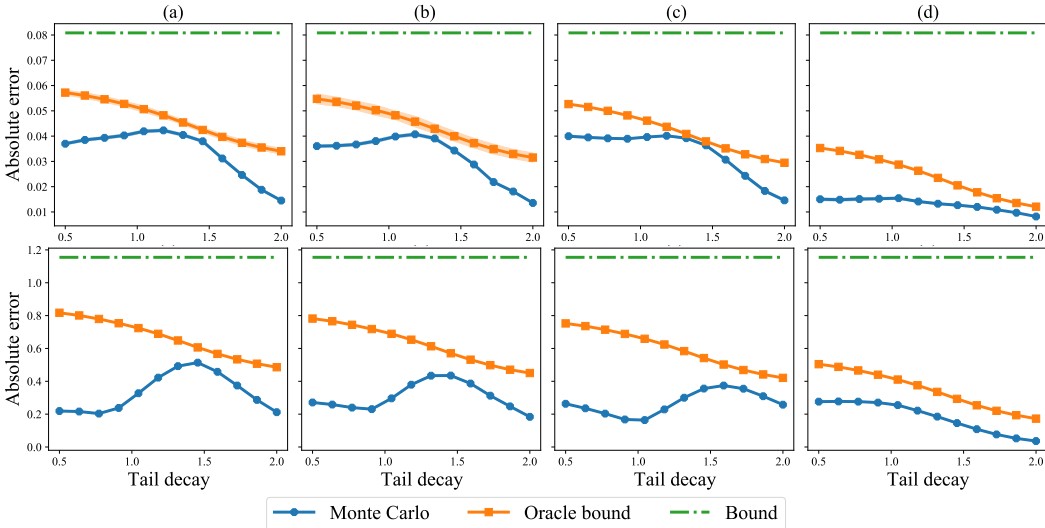

Figure 11: Absolute error versus decaying index of $Q$ on synthetic data with $k = 10^3$ and $n = 10^4$ (log-log scale) for the frontier integral **(top)** and the divergence frontier **(bottom)**. **(a)**: $P \sim \mathrm{Dir}(\mathbf{1})$; **(b)**: $P \sim \mathrm{Dir}(\mathbf{1}/2)$; **(c)**: $P \sim \mathrm{Zipf}(1)$; **(d)**: $P \sim \mathrm{Zipf}(2)$.

data-dependent and chosen as $b_a = 1/2$ if $n_a = 0$, $b_a = 1$ if $n_a = 1$ and $b_a = 3/4$ otherwise. For the *Krichevsky-Trofimov* estimator, the parameter $b \equiv 1/2$. For the *Laplace* estimator, the parameter $b \equiv 1$. See Tab. 2 for a summary.

We consider the same three experiments as for the statistical error. As shown in Fig. 12, the rate of convergence in $n$ of all estimators are similar except for some fluctuations of the Good-Turing estimator. When $P = Q$ (i.e., $\mathrm{Zipf}(1)$), the add-constant estimators outperforms the empirical measures slightly while the Good-Turing estimator performs better than the empirical measures for relatively small sample size and performs worse as the sample size increases. When one of the distribution has a fast-decaying tail (i.e., $P \sim \mathrm{Zipf}(2)$), the absolute error of the add-constant estimators are much larger than the one of empirical measures, while the Good-Turing estimator has a similar performance as empirical measures. When $P$ and $Q$ are different and do not have

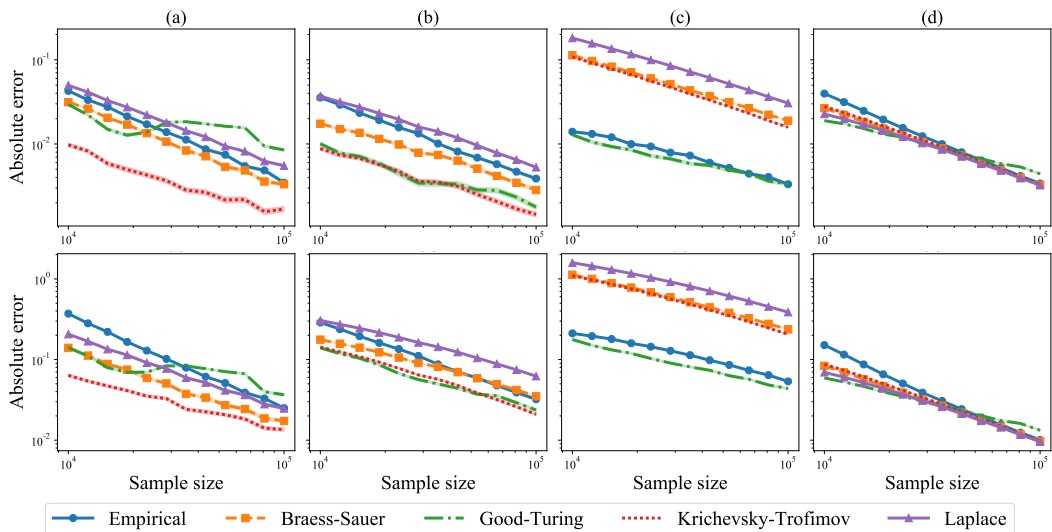

Figure 12: Statistical error versus sample size on synthetic data with $k = 10^3$ (log-log scale) for the frontier integral **(top)** and the divergence frontier **(bottom)**. **(a)**: $\mathrm{Zipf}(1)$ and Step; **(b)**: $\mathrm{Zipf}(0)$ and $\mathrm{Dir}(\mathbf{1}/2)$; **(c)**: $\mathrm{Zipf}(2)$ and $\mathrm{Dir}(\mathbf{1})$; **(d)**: $\mathrm{Zipf}(1)$ and $\mathrm{Zipf}(1)$.

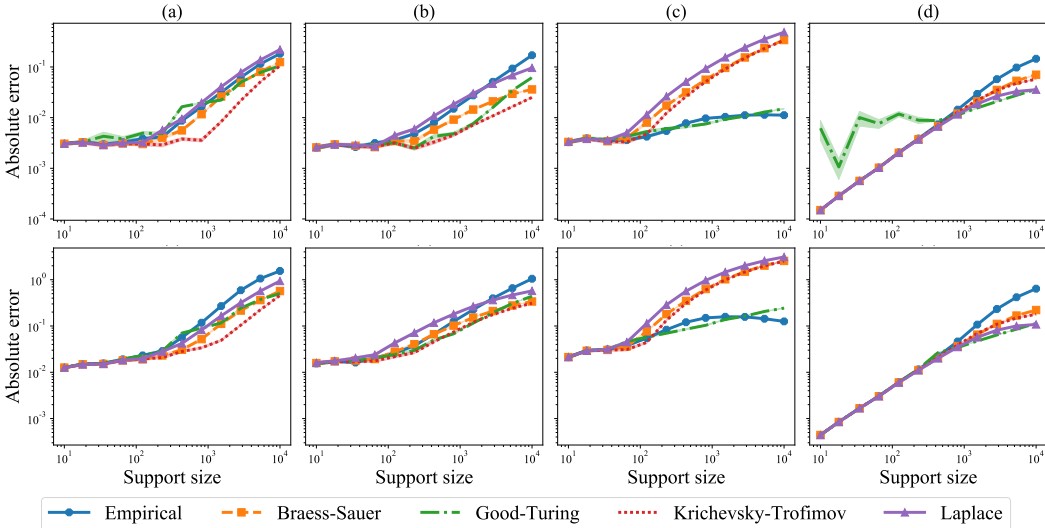

Figure 13: Absolute error versus support size on synthetic data with $n = 2 \times 10^4$ (log-log scale) for the frontier integral **(top)** and the divergence frontier **(bottom)**. **(a)**: $\mathrm{Zipf}(1)$ and Step; **(b)**: $\mathrm{Zipf}(0)$ and $\mathrm{Dir}(\mathbf{1}/2)$; **(c)**: $\mathrm{Zipf}(2)$ and $\mathrm{Dir}(\mathbf{1})$; **(d)**: $\mathrm{Zipf}(1)$ and $\mathrm{Zipf}(1)$.

fast-decaying tails, the Krichevsky-Trofimov estimator enjoys the largest improvement compared to the empirical measures.

Fig. 13 presents the results for increasing support size. The findings are similar to the ones in the first experiment except that the absolute error is increasing here rather than decreasing.

Fig. 14 shows that the Good-Turing estimator is relatively more robust to the tail decaying index than other estimators. When $P \sim \mathrm{Zipf}(2)$, the absolute error of the add-constant estimators is much larger than the one of the empirical measures in the beginning and then becomes slightly smaller in the end. In other cases, this behavior is reversed.

To summarize, when two distributions are the same, all estimators performs similarly with the Good-Turing estimator being the worst. When there is one distribution whose tail decays fast,

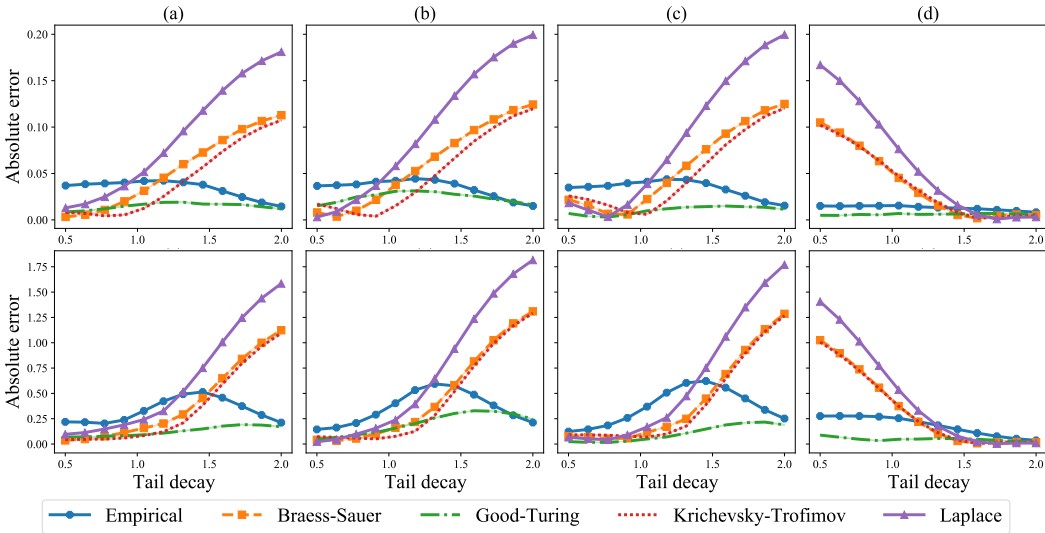

Figure 14: Absolute error versus sample size on synthetic data with $k = 10^3$ and $n = 10^4$ (log-log scale) for the frontier integral **(top)** and the divergence frontier **(bottom)**. **(a)**: $P \sim \mathrm{Dir}(\mathbf{1})$; **(b)**: $P \sim \mathrm{Step}$; **(c)**: $\mathrm{Zipf}(0)$; **(d)**: $\mathrm{Zipf}(2)$.

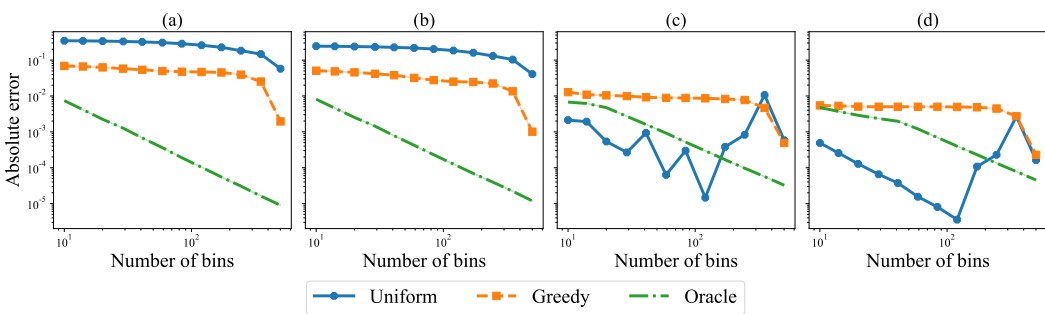

Figure 15: Absolute error versus number of bins for different quantization strategies with support size 600 (log-log scale). **(a)**: $\mathrm{Dir}(\mathbf{1})$ and $\mathrm{Dir}(\mathbf{1}/2)$; **(b)**: $\mathrm{Zipf}(0)$ and $\mathrm{Dir}(\mathbf{1}/2)$; **(c)**: $\mathrm{Zipf}(2)$ and $\mathrm{Step}$; **(d)**: $\mathrm{Zipf}(1)$ and $\mathrm{Zipf}(2)$.

the Good-Turing estimator slightly outperforms the empirical measure; while the add-constant estimators have much larger absolute errors. When the tails of both distributions decay slowly, the Krichevsky-Trofimov estimator has the best performance over all estimators.

**Quantization error.** We study the bound on the quantization error as in (6). Since the absolute error is always zero when $P = Q$, we have $21 - 6 = 15$ different pairs of $(P, Q)$. We consider three different quantization strategies: 1) the *uniform* quantization which quantizes the distributions into equally spaced bins based on their original ordering; 2) the *greedy* quantization which sorts the bins according to the ratios $\{P(a)/Q(a)\}_{a \in \mathcal{X}}$ and then add split one bin at a time so that the Frontier Integral is maximized; 3) the *oracle* quantization we used to prove (6); see also Fig. 15.

As shown in Fig. 15, the absolute error of the oracle quantization can have a faster rate than $O(k^{-1})$ in some cases. To be more specific, when both $P$ and $Q$ have slow-decaying tails, its absolute error decays roughly as $O(k^{-1.7})$; when one of them has fast-decaying tail, its absolute error decays slower than $O(k^{-1})$ in the beginning and then faster than $O(k^{-1})$. Comparing different quantization strategies, the oracle quantization always outperforms the greedy one. When either $P$ or $Q$ is not ordered, the uniform quantization has the worst performance. When both $P$ and $Q$ are ordered, its absolute error is not monotonic—it is quite small in the beginning and then becomes larger.

## G.2 Real data

We analyze the performance of the bounds as well as the various smoothed estimators in the context evaluating generative models for images and text using divergence curves. All experiments models are trained on a workstation with 8 Nvidia Quadro RTX GPUs (24G memory each). The image experiments were trained with 2 GPUs at once while the text ones used all 8.

**Tasks and datasets.** We consider two domains: images and text. For the image domain, we train a generative model for the CIFAR-10 dataset [36] based on StyleGAN2-Ada [32]. We use the publicly available code[6] with their default hyperparameters and train on 2 GPUs. In order to enable the code to run faster, we make two architectural simplifications: (a) we reduce the channel dimensions for each convolution layer in the generator from 512 to 256, and, (b) we reduce the number of styled convolution layers for each resolution from 2 to 1. In particular, the latter effectively cuts the number of convolution layers in half. This leads to a 6.6x reduction in running time at the cost of a slightly worse FID [27] of 4.7 rather than the 2.4 of the original network. In order to compute the divergence frontier, we use the test set of 10000 images as the target distribution $P$ and we sample 10000 images from the generative model as the model distribution $Q$.

For the text domain, we finetune a pretrained GPT-2 [50] model with 124M parameters (i.e., GPT-2 small) on the Wikitext-103 dataset [43]. We use the open-source HuggingFace Transformers library [65] for training. To form a sufficiently large evaluation set, we finetune on 90% of the wikitext-103 training dataset, and use the remaining 10% plus the validation set as an evaluation set. Finetuning is done on 4 GPUs for 2k iterations, with sequences of 1024 tokens and a batch size of 8 sequences. For generation, we split the evaluation set into 10k sequences of 500 tokens, and split each sequence into a prefix of length 100 and a continuation of length 400. The prefix paired with the continuation (a "completion") is considered a sample from $P$. Using the finetuned model we generate a continuation for each prefix using top-$p$ sampling with $p = 0.9$. Each prefix paired with its generated continuation is considered a sample from $Q$.

**Settings.** In order to compute the divergence frontier, we jointly quantize $P$ and $Q$, not directly in a raw image/text space, but in a feature space [54, 37, 27]. Specifically, we represent each image by its features from a pretrained ResNet-50 model [25], and each text generation by its terminal hidden state under a pretrained the 774M GPT-2 model (i.e., GPT-2 large). In order to quantize these features, we learn a 4 or 5 dimensional embedding of the image/text features using a deep network which maintains the neighborhood structure of the data while encouraging the features to be uniformly distributed on the unit sphere [53], and simply quantize these embeddings on a uniform lattice with $k$ bins. For each support size $k$, this gives us quantized distributions $P_{\mathcal{S}_k}, Q_{\mathcal{S}_k}$. We then sample $n$ i.i.d points each from these distributions and consider the empirical distributions $\hat{P}_{\mathcal{S}_k,n}, \hat{Q}_{\mathcal{S}_k,n}$ as well as the add-constant and Good-Turing estimators computed form these samples. We repeat this 100 times to a Monte Carlo estimate of the expected absolute error $\mathbb{E}|\mathrm{FI}(\hat{P}_{\mathcal{S}_k,n}, \hat{Q}_{\mathcal{S}_k,n}) - \mathrm{FI}(P_{\mathcal{S}_k}, Q_{\mathcal{S}_k})|$ as well as its standard error.

**Statistical error.** We compare the distribution-dependent bound ("oracle bound") and the distribution-free bound ("bound") to the Monte Carlo estimates described above. We consider two experiments. First, we fix the support size $k$ and vary the sample size $n$ from 100 to 25000. Second, we fix the sample size $n$ and vary the support size $k$ from 8 to 2048 in powers of 2.

We observe Fig. 16 that both the distribution-free and distribution-dependent bounds decrease with the sample size $n$ at a similar rate. For $k = 1024$ or $k = 2048$, we observe that the bound has approximately the same slope as the Monte Carlo estimate in log-log scale; this means that they exhibit a near-identical rate in $n$. On the other hand, the Monte Carlo estimates exhibit fast rates of convergence than the bound for $k = 64$ or $k = 128$. Therefore, the bounds capture the worst-case behavior of real image and text data.

Next, we see from Fig. 17 that the two bounds again exhibit near-identical rates with the support size $k$. We observe again that the slope of the Monte Carlo estimate and that of the bounds are close for $n = 1000$, indicating a similar scaling with respect to $k$. However, the Monte Carlo estimate grows faster than the bound for $n = 10000$.

---

[6] https://github.com/NVlabs/stylegan2-ada-pytorch

Table 3: The frontier integral with pretrained and finetuned feature embedding models.

| Quantization level $k$ | 2 | 4 | 8 | 16 | 32 | 64 | 128 | 256 | 512 | 1024 |
|---|---|---|---|---|---|---|---|---|---|---|
| Pretrained | 3.38e-5 | 2.64e-5 | 2.84e-4 | 6.95e-4 | 1.47e-3 | 3.25e-3 | 6.28e-3 | 1.18e-2 | 2.52e-2 | 5.09e-2 |
| Finetuned | 7.23e-6 | 1.37e-4 | 3.98e-4 | 1.77e-3 | 2.36e-3 | 5.31e-3 | 9.84e-3 | 1.95e-2 | 3.49e-2 | 6.34e-2 |

**Distribution estimators.** As in the previous section, we compare the empirical estimator, the (modified) Good-Turing estimator, and three add-$b$ smoothing estimators, namely Laplace, Krichevsky-Trofimov and Braess-Sauer. We consider the same two experiments as for the statistical error.

From Fig. 18, we see that for $n > k$, we observe similar rates (i.e., similar slopes) for all estimators with respect to the sample size $n$. The absolute error of the Good-Turing estimator is the worst among all estimators considered for $k = 64$ or $k = 128$ and $n$ large. However, for $k = 1024$ or $k = 2048$, the empirical estimator is the worst. The various add-$b$ estimators work the best in the regime of $n < k$, where each add-$b$ estimator attains the smallest error at a different $n$. In particular, the Laplace estimator is the best or close to the best in all each of the settings considered.

Fig. 19 shows the corresponding results for varying $k$. The results are similar to the previous setting, expect the error increases with $k$ rather than decreases.

**Performance across training.** Next, we visualize the divergence frontiers and the corresponding frontier integral across training in Fig. 20. On the left, we plot the divergence curve at initialization (or with the pretrained model in case of text), at the first checkpoint ("Partly") and the fully trained model ("Final"). We observe that the divergence frontiers for the fully trained model are closer to the origin than the partially trained ones or the model at initialization. This denotes a smaller loss of precision and recall for the fully trained model. The frontier integral, as a summary statistic, shows the same trend (right).

**Fine-tuning the feature embedding model.** In our real data experiments, we follow the common practice in this line of research [54, 18, 49] and use a pre-trained feature embedding model to extract feature representations. We also design a procedure to fine-tune the feature embedding model for comparing two distributions here. Concretely, we compare the frontier integral using the following two feature embedding models. First, we use a pretrained 4-layer ConvNet to extract feature embeddings for the generations of the StyleGAN. Second, we reinitialize the output layer of the 4-layer ConvNet, finetune it to distinguish true images from generated ones, and use the finetuned ConvNet to extract features. Finally, we compute the frontier integral using k-means clustering for various values of $k$. As shown in Tab. 3, the frontier integrals computed via the finetuned ConvNet are slightly larger than the ones without finetuning. This is as expected since the finetuned model usually gives a better feature representation in the sense of distinguishing distributions.

## H   Length of the divergence frontier

In this section, we discuss how the length of the divergence frontier is different from the frontier integral. In particular, we show that the length of the divergence frontier is lower bounded by the Jeffery divergence, which could be unbounded, whereas the frontier integral is always bounded between $0$ and $1$.

**Setup.** Let $P, Q$ be two distributions on a finite alphabet $\mathcal{X}$. Recall that the divergence frontier is defined as the parametric curve $\mathcal{F}(P, Q) := (x(\lambda), y(\lambda))$ for $\lambda \in (0, 1)$ where

$$
\begin{aligned}
x(\lambda) &= \mathrm{KL}_{1-\lambda}(Q\|P) = \sum_{a \in \mathcal{X}} Q(a) \log \frac{Q(a)}{\lambda P(a) + (1 - \lambda)Q(a)} \\
y(\lambda) &= \mathrm{KL}_{\lambda}(P\|Q) = \sum_{a \in \mathcal{X}} P(a) \log \frac{P(a)}{\lambda P(a) + (1 - \lambda)Q(a)} .
\end{aligned}
\tag{25}
$$

Recall that the Jeffery divergence between $P$ and $Q$ is defined as

$$
\mathrm{JD}(P, Q) = \mathrm{KL}(P\|Q) + \mathrm{KL}(Q\|P) = \sum_{a \in \mathcal{X}} \big(P(a) - Q(a)\big)\big(\log P(a) - \log Q(a)\big) .
$$

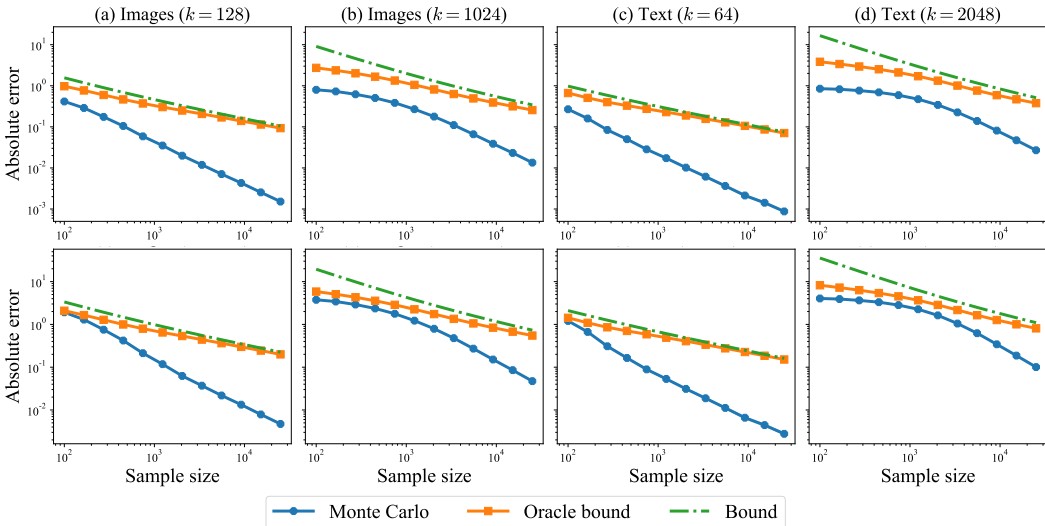

Figure 16: Absolute error versus sample size $n$ on real data (log-log scale) for the frontier integral **(top)** and the divergence frontier **(bottom)**. **Left Two**: Image data (CIFAR-10) with support sizes $k = 128$ and $k = 1024$. **Right Two**: Text data (WikiText-103) with support sizes $k = 64$ and $k = 2048$. The bounds are scaled by $15$.

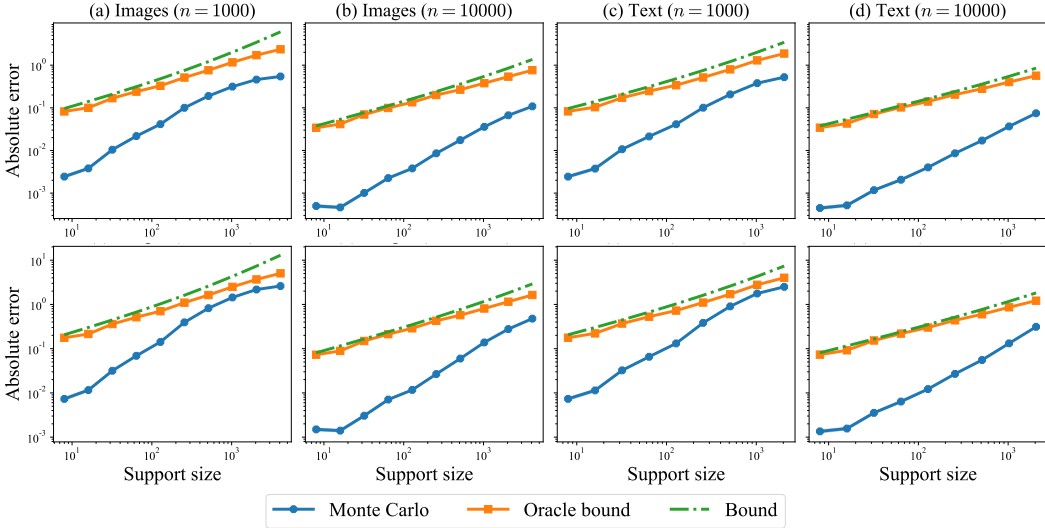

Figure 17: Absolute error versus support size $k$ on real data (log-log scale) for the frontier integral **(top)** and the divergence frontier **(bottom)**. **Left Two**: Image data (CIFAR-10) with sample sizes $n = 1000$ and $n = 10000$ **Right Two**: Text data (WikiText-103) with sample sizes $n = 1000$ and $n = 10000$. The bounds are scaled by $15$.

Note that $\mathrm{JD}(P, Q)$ is unbounded when there exists an atom such that $P(a) = 0, Q(a) \neq 0$ or $P(a) \neq 0, Q(a) = 0$.

We show that the length of the divergence frontier between $P, Q$ is lower bounded by the corresponding Jeffrey's divergence, which can be unbounded.

**Proposition 28.** *Consider two distributions $P, Q$ on a finite alphabet $\mathcal{X}$. The length $\mathrm{length}(\mathcal{F}(P, Q))$ of the divergence frontier $\mathcal{F}(P, Q)$ satisfies*

$$\mathrm{length}(\mathcal{F}(P, Q)) \geq \frac{1}{\sqrt{2}} \mathrm{JD}(P, Q).$$

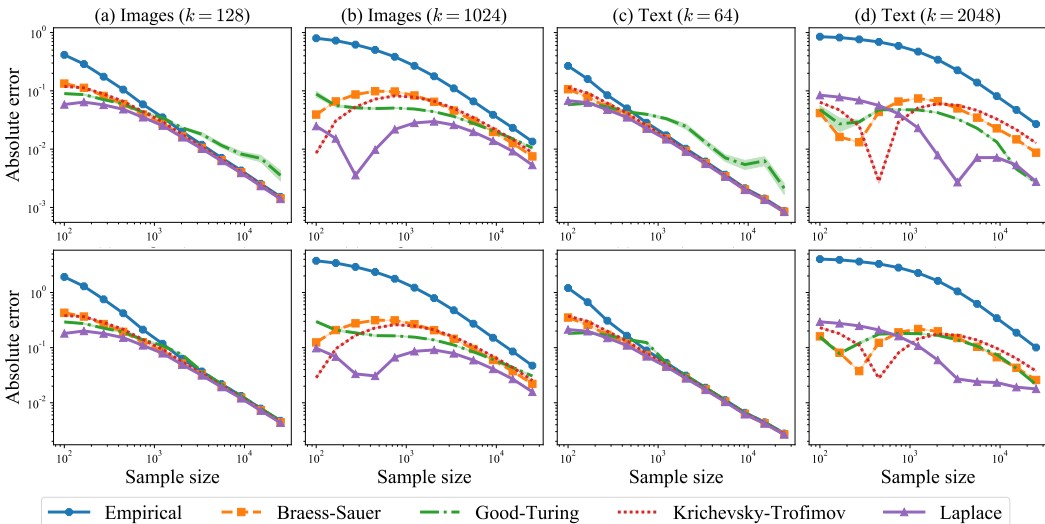

Figure 18: Absolute error versus sample size $n$ on real data (log-log scale) for the frontier integral **(top)** and the divergence frontier **(bottom)**. **Left Two**: Image data (CIFAR-10) with support size $k = 128$ and $k = 1024$ **Right Two**: Text data (WikiText-103) with support size $k = 64$ and $k = 2048$.

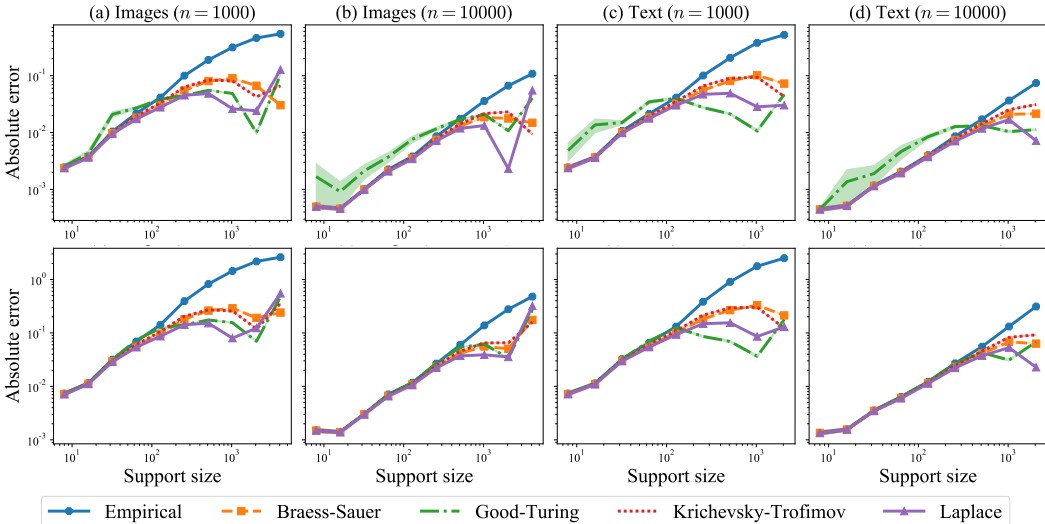

Figure 19: Absolute error versus support size $k$ on real data for the frontier integral **(top)** and the divergence frontier **(bottom)**. **Left Two**: Image data (CIFAR-10) with sample sizes $n = 1000$ and $n = 10000$. **Right Two**: Text data (WikiText-103) with sample sizes $n = 1000$ and $n = 10000$.

*Proof.* We assume without loss of generality that $P(a) + Q(a) > 0$ for each $a \in \mathcal{X}$. Define shorthand $R_\lambda = \lambda P + (1 - \lambda)Q$. We bound the length of the divergence frontier $\mathcal{F}(P, Q)$, which is given by

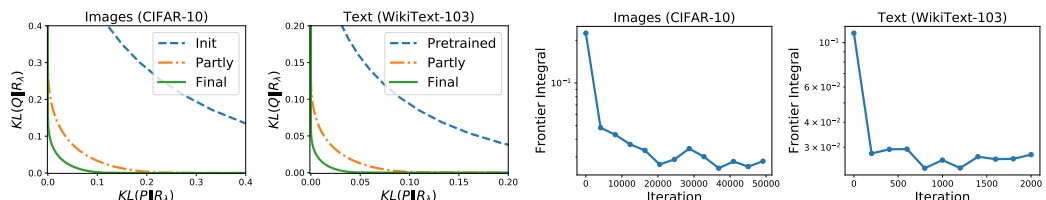

Figure 20: **Left Two**: The divergence frontier at different points in training. **Right Two**: The frontier integral plotted at different training checkpoints.

$\int_0^1 L(\lambda)\mathrm{d}\lambda$, as

$$L(\lambda)^2 = x'(\lambda)^2 + y'(\lambda)^2$$

$$= \left(\sum_{a\in\mathcal{X}} Q(a)\frac{Q(a) - P(a)}{R_\lambda(a)}\right)^2 + \left(\sum_{a\in\mathcal{X}} P(a)\frac{Q(a) - P(a)}{R_\lambda(a)}\right)^2$$

$$\geq \frac{1}{2}\left(\sum_{a\in\mathcal{X}}\frac{(P(a) - Q(a))^2}{R_\lambda(a)}\right)^2 =: \frac{1}{2}\widetilde{L}(\lambda)^2\,,$$

where we used the inequality $(a - b)^2 \leq 2(a^2 + b^2)$ for $a, b \in \mathbb{R}$. We can now complete the proof by computing this integral as

$$\sqrt{2}\cdot\mathrm{length}(\mathcal{F}(P, q)) \geq \int_0^1 \widetilde{L}(\lambda)\mathrm{d}\lambda$$

$$= \int_0^1 \sum_{a\in\mathcal{X}}\frac{(P(a) - Q(a))^2}{R_\lambda(a)}\mathrm{d}\lambda$$

$$= \sum_{a\in\mathcal{X}}(P(a) - Q(a))^2 \int_0^1 \frac{1}{\lambda P(a) + (1 - \lambda)Q(a)}\mathrm{d}\lambda$$

$$= \sum_{a\in\mathcal{X}}(P(a) - Q(a))(\log P(a) - \log Q(a)) = \mathrm{JD}(P, Q)\,.$$

$\square$

# I   Technical lemmas

We state here some technical results used in the paper.

**Theorem 29** (McDiarmid's Inequality). *Let $X_1, \cdots, X_m$ be independent random variables such that $X_i$ has range $\mathcal{X}_i$. Let $\Phi : \mathcal{X}_1 \times \cdots \times \mathcal{X}_n \to \mathbb{R}$ be any function which satisfies the bounded difference property. That is, there exist constants $B_1, \cdots, B_n > 0$ such that for every $i = 1, \cdots, n$ and $(x_1, \cdots, x_n), (x'_1, \cdots, x'_n) \in \mathcal{X}_1 \times \cdots \mathcal{X}_n$ which differ only on the $i^{th}$ coordinate (i.e., $x_j = x'_j$ for $j \neq i$), we have,*

$$|\Phi(x_1, \cdots, x_n) - \Phi(x'_1, \cdots, x'_n)| \leq B_i\,.$$

*Then, for any $t > 0$, we have,*

$$\mathbb{P}\left(|\Phi(X_1, \cdots, X_n) - \mathbb{E}[\Phi(X_1, \cdots, X_n)]| > t\right) \leq 2\exp\left(-\frac{2t^2}{\sum_{i=1}^n B_i^2}\right)\,.$$

**Property 30.** *Suppose $f : (0, \infty) \to [0, \infty)$ is convex and continuously differentiable with $f(1) = 0 = f'(1)$. Then, $f'(x) \leq 0$ for all $x \in (0, 1)$ and $f'(x) \geq 0$ for all $x \in (1, \infty)$.*

*Proof.* Monotonicity of $f'$ means that we have for any $x \in (0, 1)$ and $y \in (1, \infty)$ that $f'(x) \leq f'(1) = 0 \leq f'(y)$. $\square$

**Lemma 31.** *For all $x \in (0, 1)$ and $n \geq 3$, we have*

$$0 \leq (1-x)^n x \log \frac{1}{x} \leq \frac{\log n}{n} \, .$$

*Proof.* Let $h(x) = (1-x)^n x \log(1/x)$ be defined on $(0,1)$. Since $\lim_{x \to 0} h(x) = 0 < h(1/n)$, the global supremum does not occur as $x \to 0$. We first argue that $h$ obtains its global maximum in $(0, 1/n]$. We calculate

$$h'(x) = (1-x)^{n-1} \left( -nx \log \frac{1}{x} + (1-x) \left( \log \frac{1}{x} - 1 \right) \right) \leq (1-x)^{n-1}(1-nx) \log \frac{1}{x} \, .$$

Note that $h'(x) < 0$ for $x > 1/n$, so $h$ is strictly decreasing on $(1/n, 1)$. Therefore, it must obtain its global maximum on $(0, 1/n]$. On this interval, we have,

$$(1-x)^n x \log \frac{1}{x} \leq x \log \frac{1}{x} \leq \frac{\log n}{n} \, ,$$

since $x \log(1/x)$ is increasing on $(0, \exp(-1))$. $\qquad \square$

The next lemma comes from [3, Theorem 1].

**Lemma 32.** *For all $x \in (0, 1)$ and $n \geq 1$, we have*

$$0 \leq (1-x)^n x \leq \exp(-1)/(n+1) < 1/n \, .$$