# OpenReview forum: "Divergence Frontiers for Generative Models: Sample Complexity, Quantization Effects, and Frontier Integrals"
_NeurIPS.cc/2021/Conference — NeurIPS 2021 Poster_

### Official Review · Reviewer_cCnu · 2021-07-11

**Rating:** 5
**Confidence:** 4

**Summary:**

The paper proposes the frontier integral to measure the precision and recall for evaluate the fitted density model, and provide the complexity bound  for the quantized pluged in estimator. It also shows the smoothing technique can improve the bound.

**Limitations And Societal Impact:**

Yes, the limitations are mentioned in the paper.

**Main Review:**

Originality: The frontier integral is a natural extension to the previously proposed "divergence frontiers".

Clarity: The paper is clearly written and well organized.

Quality:
The complexity analysis is sound and the benefit of the smoothing technique is reasonable and intuitive.  I do agree the proposed measurement is valuable for low-dimensional problems.

However, if the goal is to evaluate the high-dimensional image and text generative model, my major concern is the practical usefulness of the measure.
Major:
1. The goal for the measure is to evaluate the generative model, which usually has high dimensions. The proposed quantization estimation is not very practical in high-dimensional space. Therefore, in practice, the author uses some pre-trained networks to project the original distribution to low dimensional space: 4d for images and 5d for text. Dimension 4 is much lower than the intrinsic dimension of the data/model distribution, which will largely underestimate (under-represent) the support of the original distribution, so will recall in the representation space will still reflect the recall in the original space? The performance seems to depend heavily on the feature extractor,  experiments need to be done to show that if the final evaluation will reflect the recall and precision in the original space.

2. For the GAN model which doesn't allow a density (or when the data distribution lies on a low-dimensional manifold), the proposed measure seems no longer well-defined, so is the GAN experiment still meaningful?

Minor:
The KL and RKL seem reasonable to measure the precession and recall due to their moment/mode-matching proterioes. What's the benefit to generalize to other f-divergence?


Significance: If the frontier integral is a practically significant measurement for the evaluation of the low-dimensional generative model. However, it is not clear that the proposed method is practically useful for the high-dimension model.

**Time Spent Reviewing:**

8 hours

---

> ### Author Response · Authors · 2021-08-10
> **Response to Reviewer cCnu**
>
> Thank you for your insightful and concrete comments! We address your main comments below. For better readability, we use bold face for your comments.
>
> **In practice, the author uses some pre-trained networks to project the original distribution to low dimensional space: 4d for images and 5d for text.**
>
> *TL;DR*: The use of a 4 or 5 dimensional representation is a shortcoming of the particular quantization algorithm, lattice quantization (Sablayrolles et. al. 2019), where the support size $k$ determines the dimensionality of the representation. Experiments with k-means which directly use the high dimensional embeddings from pre-trained deep nets (ResNet or GPT-2) demonstrate similar results.
>
> *Long answer*: The typical process of computing the divergence frontiers involves the steps (Sajjadi et. al. 2018, Djolonga et. al. 2019):
> * Compute dense embeddings of the images/text (we use pretrained ResNet-18 for CIFAR-10 and GPT-2 large for text) for samples from the target distribution and the model distribution;
> * Use the dense embeddings to perform a data-dependent quantization to quantize the target distribution $\hat P_n$ and model distribution $\hat Q_n$ into multinomial distributions $\hat P_{n, S}$ and $\hat Q_{n, S}$ respectively;
> * Compute the divergence frontiers between the quantized distributions.
> In our experiments, we use lattice quantization (Sablayrolles et. al. 2019) for step 2. This involves building a lower dimensional representation of the images, and the dimensionality determines the support size $k$ of the quantized distribution. *This is a shortcoming of lattice quantization*.
>
> We experiment with k-means clustering which does not have the same drawback. It works directly with the dense embeddings from ResNet/GPT-2. We obtain nearly identical results in this case. Specifically, we compute two sets of FIs by quantizing the embeddings from ResNet into $k = 8, 16, 32, \dots, 1024$ bins with k-means clustering and lattice quantization, respectively. The *spearman rank correlation* between these two sets of FIs is *0.9524*; while for GPT-2, the spearman rank correlation is *1.0*.
>
> **Experiments need to be done to show that if the final evaluation will reflect the recall and precision in the original space.**
>
> We did an *additional experiment* to show that the FI can be accurately estimated in moderate dimension if we have enough sample. Since it is infeasible to evaluate the exact FI in high dimension, we instead consider a moderate dimension $d = 10$ and compute the absolute error for the estimated FI using k-means clustering. This [figure](https://anonymous.4open.science/r/frontier-integral-rebuttal-60BA/synthetic-cont-quant-level-dim10.png) shows that the error decreases as the sample size increases, suggesting that the estimation will be accurate if we have enough samples.
> To be more specific, we 1) choose $P = N_{10}(0, I_{10})$ and $Q = N_{10}(1, I_{10})$, 2) generate $n$ samples from each of them, 3) estimate FI$(P, Q)$ from these samples using k-means clustering with $k \propto n^{1/3}$.
>
> **For the GAN model which doesn't allow a density (or when the data distribution lies on a low-dimensional manifold), the proposed measure seems no longer well-defined, so is the GAN experiment still meaningful?**
>
> Both the divergence frontier and the frontier integral work for arbitrary measures; see Appendix A for the precision definition. Note that we always have a density w.r.t. the dominating measure $\mu = (P+Q)/2$. Moreover, in practice, one only needs samples from the model distribution.
>
> **The KL and RKL seem reasonable to measure the precession and recall due to their moment/mode-matching properties. What's the benefit to generalize to other f-divergence?**
>
> We discuss $f$-divergences mainly to highlight the generality of the proof.

---

> > ### Comment · Reviewer_cCnu · 2021-08-18
> > **Major concerns remain**
> >
> > Major:
> > I think the major difficulty of evaluating the generative model is the curse of dimensionality. If there is a perfect dimension reduction method to project all the data or samples to low-dimensional space, then any two-sample test can be used. Then any distance measure can be used depends on what we care about, e.g. if I care about the tail, I can use Wasserstein, if I care about the mode I can use RKL... This is actually the easy part. There are also a lot of methods to estimate those divergences like discretization or a lot of bounds can be estimated.
> >
> > My concern is not about the quantization, but the dimensionality of the embedding space. The embedding space has to be much smaller than the data dimension, so a lot of information about the distribution must be lost in the feature extractor. So when applied the proposed method in high-dimensional problems, the quality of the measure majorly depends on the black-box feature extractor, which makes me unsatisfied.
> > Although I do agree that evaluating the high-dimensional generative models is difficult, I prefer the feature extractor can be learned by some designed criteria,  not just from a pre-trained classifier or GPT2, they are not trained for measuring the distribution difference.
> >
> >
> > Others:
> > Relative density seems ok to define the generalized f divergence, thanks.

---

> > > ### Author Response · Authors · 2021-08-20
> > > **Response to Major Concerns of Reviewer cCnu**
> > >
> > > We thank the reviewer for reading our response!
> > >
> > > We would like to clarify that **we are not introducing divergence frontiers** for generative models. **Divergence frontiers were previously introduced and studied** in several papers (Sajjadi et. al. 2018), (Kynkäänniemi et. al. 2019), and (Djolonga et. al. 2019); see (Dou et. al., 2021) for recent applications. They have been shown empirically to be useful for the evaluation of the trade-offs of generative models. Note that, in (Sajjadi et. al. 2018) and (Kynkäänniemi et. al. 2019), the samples generated by the models are **high dimensional image data**. Note also that, in both (Sajjadi et. al. 2018) and (Kynkäänniemi et. al. 2019), quantization was used in the estimation pipeline for divergence frontiers.
> > >
> > > The **frontier integral**, a quantity we introduce, is a statistical summary of the divergence frontiers. This summary enjoys nice theoretical properties and seems convenient in practice. We believe it is reasonable to compute this summary in the same manner as the divergence frontiers. In Fig. 17 in the Appendix, we plot the divergence frontier and frontier integral as we train two modern generative models StyleGAN and GPT-2 on image and text data, respectively. Both of them improve throughout training. This demonstrates that the frontier integral is also effective in evaluating deep generative models.
> > >
> > > What our paper is contributing is a **statistical analysis of divergence frontiers and its integral summary**. We are not aware of any published paper theoretically analyzing the sample complexity and the relationship with the quantization level of divergence frontiers. Our paper is the **first one** to do so. We would add that, generally speaking, the first paper on a particular topic should not reasonably be expected to address all theoretical problems at once.
> > >
> > > The theoretical results we present are **new**; we are not aware of any published paper that establishes the bounds stated in Theorems 1-3. These results establish high probability bounds characterizing the sample complexity and the quantization level. We also illustrate in our experiments, on synthetic data (Fig. 6-8) and real data (Fig. 13-14), that the theoretical scalings we establish in our bounds **align well** with empirical behavior observed in practice.
> > >
> > >
> > > **The embedding space has to be much smaller than the data dimension, so a lot of information about the distribution must be lost in the feature extractor.**
> > >
> > > For image data, the dimension of the feature embedding is **2,048**; while for text data, the dimension is **768**. We believe these **dimensions are rather high** on these domains; similar setups were used by (Heusel et. al. 2017;  Sec. 3) , (Sajjadi et. al. 2018;  Sec. 4), and (Kynkäänniemi et. al. 2019;  Sec. 2).
> > >
> > > **I prefer that the feature extractor be learned by some designed criteria, not just from a pre-trained classifier or GPT.**
> > >
> > > Sajjadi et. al. (2018) have previously observed that, when comparing generated samples in the **image space** of two different models, the generated samples of the **worse** model can be ranked **higher** than the generated samples of the **better** model. Their observation was based on compelling evidence from various evaluations [1, 2, 3]. This led them to argue in favor of comparing generated samples in the **embedding space**.
> > >
> > > Previous works on statistical evaluation of generative models used **pre-trained networks** to obtain feature embeddings (e.g., Heusel et. al. 2017, Sajjadi et. al. 2018, Kynkäänniemi et. al. 2019, and Djolonga et. al. 2019). Our theoretical framework followed that common practice. While empirically challenging common practice may be of interest in future work, this is off-subject for our paper which is of **theoretical nature** and aims at theoretically analyzing empirically motivated evaluation metrics.
> > >
> > > [1] D. Lopez-Paz and M. Oquab. Revisiting Classifier Two-Sample Tests. ICLR, 2016.
> > >
> > > [2] T. Salimans et. al. Improved Techniques for Training GANs. NeurIPS, 2016.
> > >
> > > [3] L. Theis, A. van den Oord, and M. Bethge. A note on the evaluation of generative models. ICLR, 2016.

---

> > > ### Author Response · Authors · 2021-09-01
> > > **Experiments for Different Feature Extractors**
> > >
> > > In response to your concern that **the quality of the measure majorly depends on the black-box feature extractor**, we ran another experiment to show that the frontier integral performs similarly for different feature extractors on CIFAR-10 image generation with StyleGAN.
> > >
> > > We use two feature extractors:
> > > - A ResNet outputting 2048-dimensional features.
> > > - A simple LeNet-style 4-layer ConvNet outputting 512-dimensional features.
> > >
> > > We compare frontier integral (FI) using $k$-means clustering for different values of $k$ for each feature representation.
> > >
> > > The Spearman rank correlation and Pearson correlation between these two sets of FIs are **1.000 and 0.998**, respectively, indicating that these two feature extractors give similar FIs.
> > >
> > > | Quantization level $k$ | 2 | 4 | 8 | 16 | 32 | 64 | 128 | 256 | 512 | 1024 |
> > > |--------------------------------|---|---|---|-----|-----|-----|-------|-------|------|---------|
> > > | ResNet | 7.39e-5 | 3.61e-5 | 1.60e-4 | 4.75e-4 | 7.11e-4 | 2.08e-3 | 3.96e-3 | 7.85e-3 | 1.64e-2 | 2.95e-2 |
> > > | ConvNet | 3.38e-5 | 2.64e-5 | 2.84e-4 | 6.95e-4 | 1.47e-3 | 3.25e-3 | 6.28e-3 | 1.18e-2 | 2.52e-2 | 5.09e-2 |

---

> > > > ### Comment · Reviewer_cCnu · 2021-09-01
> > > > **Thanks for the additional experiments**
> > > >
> > > > Thank the authors for the additional experiments. I do appreciate the efforts but I don't think these experiments can solve my concerns.
> > > >
> > > > The feature extractors are  trained by classification tasks, which are completely independent of the goal: comparing two distributions. For example, the experiments are done in CIFAR datasets which usually have a lot of pre-trained classification networks for providing features. But for other popular image generation datasets like CelebA or datasets without labels, how to select the feature extractor? One can argue that some pre-trained networks of Imagenet can still be used, but it will be useless for evaluating grey-scale image models. Or now my classification task is to determine if the first pixel of an image is black or not, now the features network will just ignore other information in the images, such a feature extractor can not be used for the proposed methods. These examples just show that using the feature extractor from a classifier is not principled to do a two-sample test/ comparing two distributions.
> > > >
> > > > I do acknowledge the analysis contribution in the paper, but I think if the goal is to evaluate a deep generative model, they are not very useful when using such a feature extractor from a classifier.
> > > >
> > > > Thanks

---

> > > > > ### Author Response · Authors · 2021-09-02
> > > > > **Additional Experiments for Feature Extractors Trained for Comparing Distributions**
> > > > >
> > > > > We are very happy to hear that you appreciate our theoretical contribution. We acknowledge your concerns and we have one more experiment to address it.
> > > > >
> > > > > **The pretrained feature extractors are completely independent of the goal: comparing two distributions.**
> > > > >
> > > > > We agree with the reviewer that using a feature extractor trained for comparing two distributions may give a better feature representation for the purpose of generative model evaluation. To experimentally explore this idea, we
> > > > >
> > > > > - reinitialize the output layer of the 4-layer ConvNet and finetune it to distinguish true images from generated ones;
> > > > > - use the finetuned ConvNet to extract features;
> > > > > - compute the frontier integral (FI) using k-means clustering for various k.
> > > > >
> > > > > As shown in the table below, the FIs computed via the finetuned ConvNet are *larger* than the ones without finetuning. This is consistent with the expectation that the *finetuned model gives a better feature representation* in the sense of distinguishing distributions.
> > > > >
> > > > > | Quantization level $k$ | 2 | 4 | 8 | 16 | 32 | 64 | 128 | 256 | 512 | 1024 |
> > > > > |-------------------------------|---|---|---|---|---|---|---|---|---|---|
> > > > > | Pretrained | 3.38e-5 | 2.64e-5 | 2.84e-4 | 6.95e-4 | 1.47e-3 | 3.25e-3 | 6.28e-3 | 1.18e-2 | 2.52e-2 | 5.09e-2 |
> > > > > | Finetuned | 7.23e-6 | 1.37e-4 | 3.98e-4 | 1.77e-3 | 2.36e-3 | 5.31e-3 | 9.84e-3 | 1.95e-2 | 3.49e-2 | 6.34e-2 |
> > > > >
> > > > > Thank you for making this crucial point! We will update the experiments on real data with this new approach and elaborate more on this point in the camera-ready final version.

---

### Official Review · Reviewer_qoxT · 2021-07-14

**Rating:** 6
**Confidence:** 4

**Summary:**

Inspired by the framework of divergence frontiers, the authors introduce a new "integral summary" called the frontier integral (FI, eqn (1)) and analyze its statistical properties for the KL case, and also briefly the more general f-divergence case. They experimentally show how traditional smoothing techniques (which help with the low-probability bins) affect the error in estimating the FI.

**Limitations And Societal Impact:**

The authors do not mention any (negative) societal impacts of the work, but this is primarily a theoretical contribution.

**Main Review:**

The authors start by defining the frontier integral and focus most of the paper on it. I find this confusing in that the motivation given in the abstract and the introduction is that of analyzing the frontier, but the paper mainly focuses on the quantity they have introduced. It is not clear to me how one can interpret this quantity in terms of precision / recall, or if it has any implications on them. Even if the FI is introduced solely as a theoretical gadget which would imply bounds on the precision / recall estimates, that does not seem to be the case. Just from the title I expected results that statistically compare the precision-recall curves, but I do not see anything like tha in the paper. The only connection I see is on lines 170-177, which is a result about the convergence of the KL divergence after truncation and seems like an independent result. There is further no mention of the implications of their quantization results on the estimated precision / recall. The paper could be significantly improved if the authors provide further discussion on this point. As it stands now, this makes it hard for me to evaluate the significance of the results.

The authors focus on the KL-divergence, which is absolutely fine, but then they briefly skim over possible generalization to arbitrary f-divergences (paragraph starting on line 227). If the results indeed generalize to arbitrary f-divergences, I do not understand why the authors don't present their results from that point of view. I feel that this result (if the regularity assumptions are easy to state) should be given more attention.

I find it hard how to parse Theorem 2, as it assumes the discretization satisfies a bound, which one cannot guarantee in practice. You could improve it by depending perhaps explicitly on the right hand side of (5).

The authors then move on to analyze the behaviour of smoothed estimators, in particular those based on pseudo-counts. I find this particularly useful as empty bins (in P / Q) do happen in practice and smoothing is absolutely necessary. I suggest the authors to include the values of b used in the experiments together with the corresponding names of the estimators.

The experimental results are a bit hard to understand, as the Monte-Carlo estimator is competitive in many scenarios, and there is no clear order among the methods (when we also consider those in the appendix). This is not a strong criticism, high dimensional estimation is hard, it's just that I would recommend the authors to provide some rules of thumb that practicioners can apply.

What I think would make the experimental results **really convincing** is to compute the precision-recall curves instead of the FI, and see how do the different strategies you suggest impact that. I think this should be relatively straightforward to do and would make the paper significantly stronger if improvements in FI also make the empirical precision-recall curves closer to the theoretical ones.

### Minor comments
- In l272 you claim that the bounds are divided by 100. Why is this done and what does it mean? Are you plotting bound / 100 everywhere? Should we add 2 orders of magnitude in the plots?
- I do not understand how the text dataset is created (what are the prefixes, which distributions are being compared). Could you please elaborate more?

**Time Spent Reviewing:**

5

---

> ### Author Response · Authors · 2021-08-10
> **Response to Reviewer qoxT**
>
> Thank you for your insightful and concrete comments! We address your main comments below. For better readability, we use bold face for your comments.
>
> **It is not clear to me how one can interpret the quantity they have introduced in terms of precision / recall, or if it has any implications on them.**
>
> The frontier integral is a *summary of the divergence frontier* (or precision-recall curve). Analogously to the F-score in supervised classification, which is the harmonic mean of precision and recall, the frontier integral for generative modeling is related to the weighted average of $KL_{\lambda^*}(P \Vert Q)$ (precision) and $KL_{1-\lambda^*}(Q \Vert P)$ (recall), where
> $$
> \lambda^* := \arg\min_{\lambda \in (0, 1)} \big( \lambda KL_{\lambda}(P \Vert Q) + (1 - \lambda) KL_{1-\lambda}(Q \Vert P) \big).
> $$
> Note that
> $$
> \lambda^* KL_{\lambda^*}(P \Vert Q) + (1 - \lambda^*) KL_{1-\lambda^*}(Q \Vert P) \le FI(P, Q).
> $$
> When the frontier integral is small, the weighted average $\lambda^* KL_{\lambda^*}(P \Vert Q) + (1 - \lambda^*) KL_{1-\lambda^*}(Q \Vert P)$ (weight average of precision and recall) will also be small.
>
>
> **Just from the title I expected results that statistically compare the precision-recall curves, but I do not see anything like that in the paper. The result about the convergence of the KL divergence after truncation seems like an independent result.**
>
> We have both *pointwise* convergence results for the divergence frontier (precision-recall curve) and *uniform* convergence results for the truncated frontier (curve).
>
> As we explained on lines 170-177, we have the rate of convergence for every point $\big(KL_{\lambda}(P \Vert Q), KL_{1-\lambda}(Q \Vert P))$ on the divergence frontier (or every precision-recall pair on the precision-recall curve). As a result, any linear combination of the estimated $KL_{\lambda}(P \Vert Q)$ (precision) and estimated $KL_{1-\lambda}(Q \Vert P)$ (recall) at some $\lambda$ converges to its population counterpart. In the minimax theory of hypothesis testing, where the goal is also to study two types of errors (yet different ones than the ones considered here), it is common to theoretically analyze a linear combination of the two errors. See, e.g., Sec. 1.2 (page 5) in [1] and Th. 7 in [2]. Our approach is in the same spirit, yet for generative modeling with the two errors $KL_{\lambda}(P \Vert Q)$ and $KL_{1-\lambda}(Q \Vert P)$.
>
> We want to clarify that the *truncation* is not for the KL divergence, but for the *divergence frontier* (or precision-recall curve). The precise statement is: with probability at least $1-\delta$,
> $$
> \sup_{\lambda \in [\lambda_n, 1 - \lambda_n]} \left\lVert \big(KL_{\lambda}(\hat P_n \Vert \hat Q_n), KL_{1-\lambda}(\hat Q_n \Vert \hat P_n)\big) - \big(KL_{\lambda}(P \Vert Q), KL_{1-\lambda}(Q \Vert P)\big) \right\rVert_1 \le \frac{C\log{n}}{\lambda_n} \left( \sqrt{\frac{\log{1/\delta}}n} + \sqrt{\frac{k}{n}} + \frac{k}{n} \right).
> $$
> What this result means is that if we truncate the empirical divergence (curve) within $\lambda \in [\lambda_n, 1 - \lambda_n]$ for some $\lambda_n \rightarrow 0$ slow enough, then the truncated empirical divergence (curve) converges uniformly to its population counterpart. This is a statistical result for the convergence of the empirical divergence frontiers (precision-recall curves).
>
>
> **There is further no mention of the implications of their quantization results on the estimated precision / recall.**
>
> The quantization result *Th. 25* in Appendix holds for *general $f$-divergences*, including the interpolated KL divergence. Since each point on the estimated divergence frontier (precision-recall curve) consists of two interpolated KL divergences, this theorem gives us a pointwise quantization result similar to Eq (5) for the estimated divergence frontier (precision-recall curve). We have added the discussion on this point to make it more clear in the main text.
>
>
> **I feel that the generalization to $f$-divergences (if the regularity assumptions are easy to state) should be given more attention.**
>
> Thank you for the positive feedback! We stated the regularity assumptions informally on lines 240-246. We have expanded the discussion to give it more attention in the updated version.
>
>
> **I find it hard to parse Theorem 2, as it assumes the discretization satisfies a bound, which one cannot guarantee in practice. You could improve it by depending perhaps explicitly on the right hand side of (5).**
>
> Thank you for the suggestion (we suspect you are referring to the LHS of Eq (5)). We have modified the statement of Th. 2.
>
>
> **I suggest the authors to include the values of b used in the experiments together with the corresponding names of the estimators.**
>
> Thank you for the suggestion. We have added the following table (we use cases in LaTeX for Braess-Sauer rather than multiple lines) in the updated version of the paper.
>
> | Laplace 	| Krichevsky-Trofimov | Braess-Sauer                                                                                                     	|
> |-------------|---------------------|----------------------------------------------------------------------------------------------------------------------|
> | $b\equiv 1$ | $b \equiv 1/2$  	| $b_a = 1/2$ if $a$ does not appear  |
> |  | | $b_a = 1$ if $a$ appears once |
> |  |  |   $b_a = 3/4$ if $a$ appears more than once|
>
>
> **I would recommend the authors to provide some rules of thumb that practitioners can apply.**
>
> We gave some rules of thumb on lines 887-891 in Appendix. As shown in Figures 9-11, the *Krichevsky-Trofimov* estimator ($b=1/2$) achieves the best performance in most of the cases.
>
>
> **What I think would make the experimental results really convincing is to compute the precision-recall curves instead of the FI, and see how do the different strategies you suggest impact that.**
>
> Thank you for the great suggestion! We repeated the *experiments* in Figure 15 and Figure 16 in Appendix for the *divergence frontiers* (precision-recall curves) and obtained two figures (you can find them [here](https://anonymous.4open.science/r/frontier-integral-rebuttal-60BA/real-smoothing-n-div-frontier.png) and [here](https://anonymous.4open.science/r/frontier-integral-rebuttal-60BA/real-smoothing-k-div-frontier.png)). We found that
> * Smoothed distribution estimators *improve the estimation accuracy* for the divergence frontiers (precision-recall curves).
> * The *rates of the estimation error* for the divergence frontiers (precision-recall curves) are *quite similar* to the ones for the FI in Figure 15 and Figure 16.
>
> To be more specific, the error for the divergence frontiers (precision-recall curves) is measured by the $L_\infty$ norm of the Euclidean distance between the estimated curve and the population curve, i.e.,
> $$
> \sup_{\lambda \in (0, 1)} \lVert \big( KL_{\lambda}(\hat P \Vert \hat Q), KL_{1-\lambda}(\hat Q \Vert \hat P) \big) - \big( KL_{\lambda}(P \Vert Q), KL_{1-\lambda}(Q \Vert P) \big) \rVert_2.
> $$
>
>
> **In l272 you claim that the bounds are divided by 100. Why is this done and what does it mean?**
>
> In this experiment, we aim to check whether the *rate of convergence* in the bounds reflects the true one in the Monte Carlo estimation, while the *absolute value is not as important*. In other words, the slope of the line in log-log scale is important but not the intercept. To facilitate this comparison, we divide all the bounds by a constant factor of 100 in this experiment, which maintains the same slope but changes the intercept in log-log scale (by $log(100) \approx 4.6$).
>
>
> **I do not understand how the text dataset is created (what are the prefixes, which distributions are being compared).**
>
> More details about the text dataset are in *Appendix G.2 (lines 925-929)*. We use the Wikitext-103 dataset, and fine-tune on 90% of its training split with the standard language modeling objective. We form an evaluation set with the remaining 10% of the training split plus the validation set. We did this to ensure the evaluation set was large enough for our analysis.
> For generation, we then split the evaluation set into 10k sequences of 500 tokens each, and split each sequence into a prefix of length 100 and a continuation of length 400. The prefix paired with the continuation is considered a sample from $P$. Using the fine tuned model, we generate a continuation for each prefix (top-$p$ sampling with $p=0.9$). Each prefix paired with its generated continuation is considered a sample from $Q$.
>
> **References**
>
> [1] Y. I. Ingster and I. A. Suslina. Nonparametric goodness-of-fit testing under Gaussian models. Springer, 2003.
>
> [2] T. T. Cai, X. J. Jeng and J. Jin. Optimal detection of heterogeneous and heteroscedastic mixtures. JRSS-B, 2011.

---

> > ### Comment · Reviewer_qoxT · 2021-08-24
> > **Response**
> >
> > Thank you for the response and the additional experiments! It is encouraging to see that also here you are observing benefits by smoothing. I think that if you focus on the frontiers rather than FI, the paper would be easier to understand. I understand what the FI means (the connection to F-score is on-point, note that this has been used in [45]), but I think the focus should rather be on the frontiers. Adding the rules of thumb to the main text would be also beneficial to the reader, as this is the most common conundrum one has in practice.

---

> > > ### Author Response · Authors · 2021-08-25
> > > **Thank You for Your Response**
> > >
> > > Thank you for your suggestions! These are easy edits to make. We will make these edits in the camera-ready version.

---

### Official Review · Reviewer_vaVk · 2021-07-20

**Rating:** 6
**Confidence:** 3

**Summary:**

This works studies an evaluation metric for comparing two distributions with possibly non-overlap support. The metric called "divergence frontier" was proposed in Djolonga et al. This work proved non-asymptotic bounds for quantized estimator of the metric, suggesting a good quantization level (number of bins k ~ n^{1/3}. It also suggests using smoothed empirical distribution estimators (add-constant/Good-Turing) to improve the estimation accuracy. The experiments are difficult to understand due to low clarity.

**Limitations And Societal Impact:**

Yes.

**Main Review:**

The main contribution of this paper is the theoretical analysis of empirical estimators of the "divergence frontier" metric and "frontier integral" (scalar summary of the former). The results are sensible, though I did not follow the proof in every detail.

I find the experiment section extremely difficult to understand. What is "distribution-free bound" and "distribution-dependent bound (oracle bound)"? Could you just refer to equations that appeared before? Besides, I would suggest the authors to revise the whole section and be clear about the hypothesis being checked and the conclusions drawn from the results. Currently this section only includes descriptions of Figures, mostly showing the decay of absolute error of different estimators and sample sizes. It is unclear to me how these results help justify the sample complexity in (6) and (7), nor did I see the suggested quantization level "n^{1/3}" play a role.

Another thing I find unclear is the definition frontier integral. It is unclear to me why there is a \lambda and 1 - \lambda weighting for the two terms and I have no geometric intuition about this integral. Could it be explained using Figure 1 (right)?

**Time Spent Reviewing:**

3

---

> ### Author Response · Authors · 2021-08-10
> **Response to Reviewer vaVk**
>
> Thank you for your insightful and concrete comments! We address your main comments below. For better readability, we use bold face for your comments.
>
>
> **What is distribution-free bound and distribution-dependent bound (oracle bound)?**
>
> The RHS of Eq (4) is the distribution-free bound and the LHS of Eq (4) is the oracle bound. We have made this clear in the updated version of the paper.
>
>
> **I would suggest the authors to revise the whole experiment section and be clear about the hypothesis being checked and the conclusions drawn from the results.**
>
> Thank you for the suggestion. We have updated the experiments section and made the hypotheses and conclusions more clear.
>
>
> **How these results help justify the sample complexity in (6) and (7).**
>
> These results show that the bounds in (6) and (7) correctly *capture the actual rate of convergence*. For instance, in Figure 3(a), the slopes of the two bounds are close to the slope of the Monte Carlo estimate (note that the exponent of $n$ is the slope of the line in log-log scale). This suggests that, when the quantization level $k$ is fixed, the true rate of convergence in $n$ of the statistical error is close to $n^{-1/2}$, which is what we prove in (6).
>
>
> **I did not see the suggested quantization level "n^{1/3}" play a role in these results.**
>
> We ran *additional experiments* to illustrate the quantization level $n^{1/3}$ as the reviewer requested. For $P = N_2(1, I_2)$ and $Q = N_2(0, I_2)$, we generate $n$ observations from each of them and estimate the FI by k-means clustering with $k \propto n^{1/r}$ for different values of $r$. As before, we estimate the absolute error by the Monte Carlo estimation. As shown in this [figure](https://anonymous.4open.science/r/frontier-integral-rebuttal-60BA/synthetic-cont-quant-level.png), the quantization level $n^{1/3}$ indeed gives the *fastest decay in absolute error*. This empirically validates our theoretical results.
>
>
> **Intuition on the definition of the frontier integral.**
>
> The frontier integral is an *integral summary of the linearized objective* defining the frontier.
>
> The divergence frontier is given by points of the form $(KL(P \Vert R_\lambda), KL(Q \Vert R_\lambda))$ for $\lambda \in (0, 1)$, where $R_\lambda$ is the minimizer of the linearized objective (cf. Proposition 1 and 2 in Djolonga et. al. 2019):
> $$
> F_\lambda(R; P, Q) =  \lambda \text{KL}(P \Vert R) + (1-\lambda) \text{KL}(Q \Vert R) .
> $$
> The frontier integral is simply an integral summary of the minimal linearized objective $ \min_R F_\lambda(R; P, Q)$, i.e.,
> $$
> FI(P, Q) =2 \int_0^1 \min_R F_\lambda(R; P, Q) d \lambda  =  2 \int_0^1 F_\lambda(R_\lambda; P, Q) d \lambda .
> $$

---

> > ### Comment · Reviewer_vaVk · 2021-08-27
> > **Response after rebuttal**
> >
> > Thanks for the response and additional experiment. It addressed my concern. I raised my score to 6.

---

> > > ### Author Response · Authors · 2021-08-30
> > > **Thank you for your response**
> > >
> > > Thank you for your time reading our response! We are happy that your concerns are addressed.

---

### Author Response · Authors · 2021-08-10
**Highlights of the Response**

We thank all reviewers for their hard work reviewing our paper and providing insightful and concrete comments. We included new experimental results in response to these comments and suggestions. We highlight them here and encourage the reviewers to take a look.

1. We included a figure for the absolute error of the FI versus the sample size, where the FI is estimated by the $k$-means clustering with $k \propto n^{1/r}$. Among $r \in \\{2, 3, 4, 5\\}$, the quantization level $n^{1/3}$ ($r=3$) recommended by our theoretical analysis (Th. 2) indeed gives the *fastest decay in absolute error*. You can find the figure [here](https://anonymous.4open.science/r/frontier-integral-rebuttal-60BA/synthetic-cont-quant-level.png). Please see the response to **Reviewer vaVK** for more details.

2. We included two figures repeating the experiments in Fig. 15 and Fig. 16 in Appendix yet for the divergence frontier (precision-recall curve). We found that 1) smoothed distribution estimators (e.g., add-constant) *improve the estimation accuracy* and 2) the rates of convergence are *quite similar* to the ones for FI in Fig. 15 and Fig. 16. You can find the figures [here](https://anonymous.4open.science/r/frontier-integral-rebuttal-60BA/real-smoothing-n-div-frontier.png) and [here](https://anonymous.4open.science/r/frontier-integral-rebuttal-60BA/real-smoothing-k-div-frontier.png). Please see the response to **Reviewer qoxT** for more details.

3. To show that the estimation of FIs does not heavily depend on the quantization method, we experimented with lattice quantization and k-means clustering. The spearman rank correlation is 0.9524 on the image data and 1.0 on the text data, indicating a strong correlation between the FIs obtained by the two quantization approaches. Please see the response to **Reviewer cCnu** for more details.

---

### Decision · Program_Chairs · 2021-09-27

**Decision:**

Accept (Poster)

**Comment:**

The submission proposes a new integral summary, frontier integral, for generative model evaluation, extending on recent work on divergence frontiers. All reviewers expressed concern about high dimensions. However, the theoretical contributions are of interest to the community, the authors have addressed most concerns in the reviews, and the high-dimension issue is a limitation of most related work and not just this paper. Overall, good paper. Accept!